# GenDICE: Generalized Offline Estimation of Stationary Values

**Ruiyi Zhang**[1][*][†]**, Bo Dai**[2*]**, Lihong Li**[2]**, Dale Schuurmans**[2]
[1]Duke University, [2]Google Research, Brain Team

## Abstract

An important problem that arises in reinforcement learning and Monte Carlo methods is estimating quantities defined by the stationary distribution of a Markov chain. In many real-world applications, access to the underlying transition operator is limited to a fixed set of data that has already been collected, without additional interaction with the environment being available. We show that consistent estimation remains possible in this challenging scenario, and that effective estimation can still be achieved in important applications. Our approach is based on estimating a ratio that corrects for the discrepancy between the stationary and empirical distributions, derived from fundamental properties of the stationary distribution, and exploiting constraint reformulations based on variational divergence minimization. The resulting algorithm, GenDICE, is straightforward and effective. We prove its consistency under general conditions, provide an error analysis, and demonstrate strong empirical performance on benchmark problems, including off-line PageRank and off-policy policy evaluation.

## 1 Introduction

Estimation of quantities defined by the stationary distribution of a Markov chain lies at the heart of many scientific and engineering problems. Famously, the steady-state distribution of a random walk on the World Wide Web provides the foundation of the PageRank algorithm (Langville & Meyer, 2004). In many areas of machine learning, Markov chain Monte Carlo (MCMC) methods are used to conduct approximate Bayesian inference by considering Markov chains whose equilibrium distribution is a desired posterior (Andrieu et al., 2002). An example from engineering is queueing theory, where the queue lengths and waiting time under the limiting distribution have been extensively studied (Gross et al., 2018). As we will also see below, stationary distribution quantities are of fundamental importance in reinforcement learning (RL) (e.g., Tsitsiklis & Van Roy, 1997).

Classical algorithms for estimating stationary distribution quantities rely on the ability to sample next states from the current state *by directly interacting with the environment* (as in on-line RL or MCMC), or even require the transition probability distribution to be given explicitly (as in PageRank). Unfortunately, these classical approaches are inapplicable when direct access to the environment is not available, which is often the case in practice. There are many practical scenarios where a collection of sampled trajectories is available, having been collected off-line by an external mechanism that chose states and recorded the subsequent next states. Given such data, we still wish to estimate a stationary quantity. One important example is off-policy policy evaluation in RL, where we wish to estimate the value of a policy different from that used to collect experience. Another example is off-line PageRank (OPR), where we seek to estimate the relative importance of webpages given a sample of the web graph.

Motivated by the importance of these off-line scenarios, and by the inapplicability of classical methods, we study the problem of *off-line estimation of stationary values* via a *stationary distribution corrector*. Instead of having access to the transition probabilities or a next-state sampler, we assume only access to a *fixed* sample of state transitions, where states have been sampled from an unknown distribution and next-states are sampled according to the Markov chain's transition operator. The

---

[*]Equal contribution.
[†]Work done while interning at Google.

off-line setting is indeed more challenging than its more traditional on-line counterpart, given that one must infer an asymptotic quantity from finite data. Nevertheless, we develop techniques that still allow consistent estimation under general conditions, and provide effective estimates in practice. The main contributions of this work are:

- We formalize the problem of off-line estimation of stationary quantities, which captures a wide range of practical applications.

- We propose a novel stationary distribution estimator, GenDICE, for this task. The resulting algorithm is based on a new dual embedding formulation for divergence minimization, with a carefully designed mechanism that explicitly eliminates degenerate solutions.

- We theoretically establish consistency and other statistical properties of GenDICE, and empirically demonstrate that it achieves significant improvements on several behavior-agnostic off-policy evaluation benchmarks and an off-line version of PageRank.

The methods we develop in this paper fundamentally extend recent work in off-policy policy evaluation (Liu et al., 2018; Nachum et al., 2019) by introducing a new formulation that leads to a more general, and as we will show, more effective estimation method.

## 2 BACKGROUND

We first introduce off-line PageRank (OPR) and off-policy policy evaluation (OPE) as two motivating domains, where the goal is to estimate stationary quantities given only off-line access to a set of sampled transitions from an environment.

**Off-line PageRank (OPR)**   The celebrated PageRank algorithm (Page et al., 1999) defines the ranking of a web page in terms of its asymptotic visitation probability under a random walk on the (augmented) directed graph specified by the hyperlinks. If we denote the World Wide Web by a directed graph $G = (V, E)$ with vertices (web pages) $v \in V$ and edges (hyperlinks) $(v, u) \in E$, PageRank considers the random walk defined by the Markov transition operator $v \to u$:

$$\mathbf{P}\left(u|v\right) = \frac{(1-\eta)}{|v|}\mathbf{1}_{(v,u)\in E} + \frac{\eta}{|V|}, \tag{1}$$

where $|v|$ denotes the out-degree of vertex $v$ and $\eta \in [0, 1)$ is a probability of "teleporting" to any page uniformly. Define $d_t\left(v\right) := \mathbb{P}\left(s_t = v|s_0 \sim \mu_0, \forall i < t, s_{i+1} \sim \mathbf{P}(\cdot|s_i)\right)$, where $\mu_0$ is the initial distribution over vertices, then the original PageRank algorithm explicitly iterates for the limit

$$d\left(v\right) := \begin{cases} \lim_{t\to\infty} d_t\left(v\right) & \text{if } \gamma = 1 \\ (1-\gamma)\sum_{t=0}^{\infty} \gamma^t d_t\left(v\right) & \text{if } \gamma \in (0,1). \end{cases} \tag{2}$$

The classical version of this problem is solved by tabular methods that simulate Equation 1. However, we are interested in a more scalable off-line version of the problem where the transition model is not explicitly given. Instead, consider estimating the rank of a particular web page $v'$ from a large web graph, given only a sample $\mathcal{D} = \{(v, u)_i\}_{i=1}^{N}$ from a random walk on $G$ as specified above. We would still like to estimate $d(v')$ based on this data. First, note that if one knew the distribution $p$ by which any vertex $v$ appeared in $\mathcal{D}$, the target quantity could be re-expressed by a simple importance ratio $d\left(v'\right) = \mathbb{E}_{v\sim p}\left[\frac{d(v)}{p(v)}\mathbf{1}_{v=v'}\right]$. Therefore, if one had the correction ratio function $\tau\left(v\right) = \frac{d(v)}{p(v)}$, an estimate of $d\left(v'\right)$ can easily be recovered via $d\left(v'\right) \approx \hat{p}\left(v'\right)\tau\left(v'\right)$, where $\hat{p}\left(v'\right)$ is the empirical probability of $v'$ estimated from $\mathcal{D}$. The main attack on the problem we investigate is to recover a good estimate of the ratio function $\tau$.

**Policy Evaluation**   An important generalization of this stationary value estimation problem arises in RL in the form of policy evaluation. Consider a Markov Decision Process (MDP) $\mathcal{M} = \langle S, A, \mathbf{P}, R, \gamma, \mu_0 \rangle$ (Puterman, 2014), where $S$ is a state space, $A$ is an action space, $\mathbf{P}\left(s'|s, a\right)$ denotes the transition dynamics, $R$ is a reward function, $\gamma \in (0, 1]$ is a discounted factor, and $\mu_0$ is the initial state distribution. Given a policy, which chooses actions in any state $s$ according to the probability distribution $\pi(\cdot|s)$, a trajectory $\beta = (s_0, a_0, r_0, s_1, a_1, r_1, \ldots)$ is generated by first sampling the initial state $s_0 \sim \mu_0$, and then for $t \geq 0$, $a_t \sim \pi(\cdot|s_t)$, $r_t \sim R(s_t, a_t)$, and $s_{t+1} \sim \mathbf{P}(\cdot|s_t, a_t)$. The value of a policy $\pi$ is the expected per-step reward defined as:

$$\text{Average: } \mathcal{R}(\pi) := \lim_{T\to\infty} \frac{1}{T+1}\mathbb{E}\left[\sum_{t=0}^{T} r_t\right], \quad \text{Discounted: } \mathcal{R}_\gamma(\pi) := (1-\gamma)\mathbb{E}\left[\sum_{t=0}^{\infty} \gamma^t r_t\right]. \tag{3}$$

In the above, the expectation is taken with respect to the randomness in the state-action pair $\mathbf{P}\left(s'|s, a\right)\pi\left(a'|s'\right)$ and the reward $R\left(s_t, a_t\right)$. Without loss of generality, we assume the limit exists for the average case, and hence $\mathcal{R}(\pi)$ is finite.

**Behavior-agnostic Off-Policy Evaluation (OPE)**   An important setting of policy evaluation that often arises in practice is to estimate $\mathcal{R}_\gamma(\pi)$ or $\mathcal{R}(\pi)$ given a fixed dataset $\mathcal{D} = \{(s, a, r, s')_i\}_{i=1}^N \sim \mathbf{P}(s'|s,a) p(s,a)$, where $p(s,a)$ is an unknown distribution induced by multiple unknown behavior policies. This problem is different from the classical form of OPE, where it is assumed that a known behavior policy $\pi_b$ is used to collect transitions. In the behavior-agnostic scenario, however, typical importance sampling (IS) estimators (e.g., Precup et al., 2000) do not apply. Even if one can assume $\mathcal{D}$ consists of trajectories where the behavior policy can be estimated from data, it is known that that straightforward IS estimators suffer a variance exponential in the trajectory length, known as the "curse of horizon" (Jiang & Li, 2016; Liu et al., 2018).

Let $d_t^\pi(s,a) = \mathbb{P}(s_t = s, a_t = a | s_0 \sim \mu_0, \forall i < t, a_i \sim \pi(\cdot|s_i), s_{i+1} \sim \mathbf{P}(\cdot|s_i, a_i))$. The stationary distribution can then be defined as

$$\mu_\gamma^\pi(s,a) := \begin{cases} \lim_{T\to\infty} \frac{1}{T+1} \sum_{t=0}^T d_t^\pi(s,a) = \lim_{t\to\infty} d_t^\pi(s,a) & \text{if } \gamma = 1 \\ (1-\gamma) \sum_{t=0}^\infty \gamma^t d_t^\pi(s,a) & \text{if } \gamma \in (0,1). \end{cases} \tag{4}$$

With this definition, $\mathcal{R}(\pi)$ and $\mathcal{R}_\gamma(\pi)$ can be equivalently re-expressed as

$$\mathcal{R}_\gamma(\pi) := \mathbb{E}_{\mu_\gamma^\pi}[R(s,a)] = \mathbb{E}_p\left[\frac{\mu_\gamma^\pi(s,a)}{p(s,a)} R(s,a)\right]. \tag{5}$$

Here we see once again that if we had the correction ratio function $\tau(s,a) = \frac{\mu_\gamma^\pi(s,a)}{p(s,a)}$ a straightforward estimate of $\mathcal{R}_\gamma(\pi)$ could be recovered via $\mathcal{R}_\gamma(\pi) \approx \mathbb{E}_{\hat{p}}[\tau(s,a) R(s,a)]$, where $\hat{p}(s,a)$ is an empirical estimate of $p(s,a)$. In this way, the behavior-agnostic OPE problem can be reduced to estimating the correction ratio function $\tau$, as above.

We note that Liu et al. (2018) and Nachum et al. (2019) also exploit Equation 5 to reduce OPE to stationary distribution correction, but these prior works are distinct from the current proposal in different ways. First, the inverse propensity score (IPS) method of Liu et al. (2018) assumes the transitions are sampled from a *single* behavior policy, which must be *known* beforehand; hence that approach is not applicable in behavior-agnostic OPE setting. Second, the recent DualDICE algorithm (Nachum et al., 2019) is also a behavior-agnostic OPE estimator, but its derivation relies on a *change-of-variable* trick that is only valid for $\gamma < 1$. This previous formulation becomes unstable when $\gamma \to 1$, as shown in Section 6 and Appendix E. The behavior-agnostic OPE estimator we derive below in Section 3 is applicable both when $\gamma = 1$ and $\gamma \in (0,1)$. This connection is why we name the new estimator GenDICE, for *GENeralized stationary DIstribution Correction Estimation*.

## 3   GenDICE

As noted, there are important estimation problems in the Markov chain and MDP settings that can be recast as estimating a stationary distribution correction ratio. We first outline the conditions that characterize the correction ratio function $\tau$, upon which we construct the objective for the GenDICE estimator, and design efficient algorithm for optimization. We will develop our approach for the more general MDP setting, with the understanding that all the methods and results can be easily specialized to the Markov chain setting.

### 3.1   Estimating Stationary Distribution Correction

The stationary distribution $\mu_\gamma^\pi$ defined in Equation 4 can also be characterized via

$$\mu(s',a') = \underbrace{(1-\gamma)\mu_0(s')\pi(a'|s') + \gamma \int \pi(a'|s')\mathbf{P}(s'|s,a)\mu(s,a)\,ds\,da}_{(\mathcal{T}\circ\mu)(s',a')}, \ \forall (s',a') \in S \times A. \tag{6}$$

At first glance, this equation shares a superficial similarity to the Bellman equation, but there is a fundamental difference. The Bellman operator recursively integrates out future $(s',a')$ pairs to characterize a current pair $(s,a)$ value, whereas the distribution operator $\mathcal{T}$ defined in Equation 6 operates in the reverse temporal direction.

When $\gamma < 1$, Equation 6 always has a fixed-point solution. For $\gamma = 1$, in the discrete case, the fixed-point exists as long as $\mathcal{T}$ is ergodic; in the continuous case, the conditions for fixed-point existence become more complicated (Meyn & Tweedie, 2012) and beyond the scope of this paper.

The development below is based on a divergence $D$ and the following default assumption.

**Assumption 1 (Markov chain regularity)** *For the given target policy $\pi$, the resulting state-action transition operator $\mathcal{T}$ has a unique stationary distribution $\mu$ that satisfies $D(\mathcal{T}\circ\mu\|\mu) = 0$.*

In the behavior-agnostic setting we consider, one does not have direct access to $\mathbf{P}$ for element-wise evaluation or sampling, but instead is given a fixed set of samples from $\mathbf{P}(s'|s,a)\,p(s,a)$ with respect to some distribution $p(s,a)$ over $S \times A$. Define $\mathcal{T}^p_{\gamma,\mu_0}$ to be a mixture of $\mu_0\pi$ and $\mathcal{T}_p$; *i.e.*, let

$$\mathcal{T}^p_{\gamma,\mu_0}\left(\left(s',a'\right),(s,a)\right) := (1-\gamma)\,\mu_0\left(s'\right)\pi\left(a'|s'\right) + \gamma\,\underbrace{\pi\left(a'|s'\right)\mathbf{P}\left(s'|s,a\right)p\left(s,a\right)}_{\mathcal{T}_p\left(\left(s',a'\right),(s,a)\right)}. \qquad (7)$$

Obviously, conditioning on $(s,a,s')$ one could easily sample $a' \sim \pi\left(a'|s'\right)$ to form $(s,a,s',a') \sim \mathcal{T}_p\left(\left(s',a'\right),(s,a)\right)$; similarly, a sample $(s',a') \sim \mu_0\left(s'\right)\pi\left(a'|s'\right)$ could be formed from $s'$. Mixing such samples with probability $\gamma$ and $1-\gamma$ respectively yields a sample $(s,a,s',a') \sim \mathcal{T}^p_{\gamma,\mu_0}\left(\left(s',a'\right),(s,a)\right)$. Based on these observations, the stationary condition for the ratio from Equation 6 can be re-expressed in terms of $\mathcal{T}^p_{\gamma,\mu_0}$ as

$$p\left(s',a'\right)\tau^*\left(s',a'\right) = \underbrace{(1-\gamma)\,\mu_0\left(s'\right)\pi\left(a'|s'\right) + \gamma\int\pi\left(a'|s'\right)\mathbf{P}\left(s'|s,a\right)p\left(s,a\right)\tau^*\left(s,a\right)ds\,da}_{\left(\mathcal{T}^p_{\gamma,\mu_0}\circ\tau^*\right)\left(s',a'\right)}, \qquad (8)$$

where $\tau^*\left(s,a\right) := \frac{\mu(s,a)}{p(s,a)}$ is the correction ratio function we seek to estimate. One natural approach to estimating $\tau^*$ is to match the LHS and RHS of Equation 8 with respect to some divergence $D\left(\cdot\|\cdot\right)$ over the empirical samples. That is, we consider estimating $\tau^*$ by solving the optimization problem

$$\min_{\tau \geq 0}\; D\left(\mathcal{T}^p_{\gamma,\mu_0}\circ\tau\|p\cdot\tau\right). \qquad (9)$$

Although this forms the basis of our approach, there are two severe issues with this naive formulation that first need to be rectified:

**i) Degenerate solutions:** When $\gamma = 1$, the operator $\mathcal{T}^p_{\gamma=1,\mu_0}$ is invariant to constant rescaling: if $\tau^* = \mathcal{T}^p_{\gamma=1,\mu_0}\circ\tau^*$ then $c\tau^* = \mathcal{T}^p_{\gamma=1,\mu_0}\circ\left(c\tau^*\right)$ for any $c \geq 0$. Therefore, simply minimizing the divergence $D\left(\mathcal{T}^p_{\gamma=1,\mu_0}\circ\tau\|p\cdot\tau\right)$ cannot provide a desirable estimate of $\tau^*$. In fact, in this case the trivial solution $\tau^*\left(s,a\right) = 0$ cannot be eliminated.

**ii) Intractable objective:** The divergence $D\left(\mathcal{T}^p_{\gamma,\mu_0}\circ\tau\|p\cdot\tau\right)$ involves the computation of $\mathcal{T}^p_{\gamma,\mu_0}\circ\tau$, which in general involves an intractable integral. Thus, evaluation of the exact objective is intractable, and neglects the assumption that we only have access to samples from $\mathcal{T}^p_{\gamma,\mu_0}$ and are not able to evaluate it at arbitrary points.

We address each of these two issues in a principled manner.

### 3.2 Eliminating degenerate solutions

To avoid degenerate solutions when $\gamma = 1$, we ensure that the solution is a proper density ratio; that is, the property $\tau \in \Xi := \{\tau\left(\cdot\right) \geq 0, \mathbb{E}_p\left[\tau\right] = 1\}$ must be true of any $\tau$ that is a ratio of some density to $p$. This provides an additional constraint that we add to the optimization formulation

$$\min_{\tau \geq 0}\; D\left(\mathcal{T}^p_{\gamma,\mu_0}\circ\tau\|p\cdot\tau\right), \quad \text{s.t.,} \quad \mathbb{E}_p\left[\tau\right] = 1. \qquad (10)$$

With this additional constraint, it is obvious that the trivial solution $\tau\left(s,a\right) = 0$ is eliminated as an infeasible point of Eqn (10), along with other degenerate solutions $\tau\left(s,a\right) = c\tau^*\left(s,a\right)$ with $c \neq 1$.

Unfortunately, exactly solving an optimization with expectation constraints is very complicated in general (Lan & Zhou, 2016), particularly given a nonlinear parameterization for $\tau$. The penalty method (Luenberger & Ye, 2015) provides a much simpler alternative, where a sequence of regularized problems are solved

$$\min_{\tau \geq 0}\; J\left(\tau\right) := D\left(\mathcal{T}^p_{\gamma,\mu_0}\circ\tau\|p\cdot\tau\right) + \tfrac{\lambda}{2}\left(\mathbb{E}_p\left[\tau\right] - 1\right)^2, \qquad (11)$$

with $\lambda$ increasing. The drawback of the penalty method is that it generally requires $\lambda \to \infty$ to ensure the strict feasibility, which is still impractical, especially in stochastic gradient descent. The infinite $\lambda$ may induce *unbounded variance* in the gradient estimator, and thus, *divergence* in optimization. However, by exploiting the special structure of the solution sets to Equation 11, we can show that, remarkably, it is unnecessary to increase $\lambda$.

**Theorem 1** *For $\gamma \in (0,1]$ and any $\lambda > 0$, the solution to Equation 11 is given by* $\tau^*\left(s,a\right) = \frac{u(s,a)}{p(s,a)}$.

The detailed proof for Theorem 1 is given in Appendix A.1. By Theorem 1, we can estimate the desired correction ratio function $\tau^*$ by solving only one optimization with an arbitrary $\lambda > 0$.

### 3.3 EXPLOITING DUAL EMBEDDING

The optimization in Equation 11 involves the integrals $\left(\mathcal{T}_{\gamma,\mu_0}^p \circ \tau\right)$ and $\mathbb{E}_p\left[\tau\right]$ inside nonlinear loss functions, hence appears difficult to solve. Moreover, obtaining unbiased gradients with a naive approach requires double sampling (Baird, 1995). Instead, we bypass both difficulties by applying a dual embedding technique (Dai et al., 2016; 2018). In particular, we assume the divergence $D$ is in the form of an $f$-divergence (Nowozin et al., 2016)

$$D_\phi\left(\left(\mathcal{T}_{\gamma,\mu_0}^p \circ \tau\right) \| p \cdot \tau\right) := \int p \cdot \tau\left(s,a\right) \phi\left(\frac{(\mathcal{T}_{\gamma,\mu_0}^p \circ \tau)(s,a)}{p \cdot \tau(s,a)}\right) ds\, da$$

where $\phi\left(\cdot\right) : \mathbb{R}_+ \to \mathbb{R}$ is a convex, lower-semicontinuous function with $\phi\left(1\right) = 0$. Plugging this into $J\left(\tau\right)$ in Equation 11 we can easily check the convexity of the objective

**Theorem 2** *For an $f$-divergence with valid $\phi$ defining $D_\phi$, the objective $J\left(\tau\right)$ is convex w.r.t. $\tau$.*

The detailed proof is provided in Appendix A.2. Recall that a suitable convex function can be represented as $\phi\left(x\right) = \max_f x \cdot f - \phi^*\left(f\right)$, where $\phi^*$ is the Fenchel conjugate of $\phi\left(\cdot\right)$. In particular, we have the representation $\frac{1}{2}x^2 = \max_u ux - \frac{1}{2}u^2$, which allows us to re-express the objective as

$$J\left(\tau\right) = \int p \cdot \tau\left(s',a'\right) \left\{ \max_f \left[ \frac{(\mathcal{T}_{\gamma,\mu_0}^p \circ \tau)(s',a')}{p \cdot \tau(s',a')} f - \phi^*\left(f\right) \right] \right\} ds'da' + \lambda \left\{ \max_u \left[ u\left(\mathbb{E}_p\left[\tau\right] - 1\right) - \frac{u^2}{2} \right] \right\}. \quad (12)$$

Applying the interchangeability principle (Shapiro et al., 2014; Dai et al., 2016), one can replace the inner $\max$ in the first term over scalar $f$ to maximize over a function $f\left(\cdot,\cdot\right) : S \times A \to \mathbb{R}$

$$\min_{\tau \geq 0} \max_{f:S\times A\to\mathbb{R}, u\in\mathbb{R}} J\left(\tau,u,f\right) = \left(1-\gamma\right) \mathbb{E}_{\mu_0\pi}\left[f\left(s,a\right)\right] + \gamma\mathbb{E}_{\mathcal{T}_p}\left[\tau\left(s,a\right) f\left(s',a'\right)\right]$$

$$- \mathbb{E}_p\left[\tau\left(s,a\right) \phi^*\left(f\left(s,a\right)\right)\right] + \lambda\left(\mathbb{E}_p\left[u\tau\left(s,a\right) - u\right] - \frac{u^2}{2}\right). \quad (13)$$

This yields the main optimization formulation, which avoids the aforementioned difficulties and is well-suited for practical optimization as discussed in Section 3.4.

**Remark (Other divergences):** In addition to $f$-divergence, the proposed estimator Equation 11 is compatible with other divergences, such as the integral probability metrics (IPM) (Müller, 1997; Sriperumbudur et al., 2009), while retaining consistency. Based on the definition of the IPM, these divergences directly lead to $\min$-$\max$ optimizations similar to Equation 13 with the identity function as $\phi^*\left(\cdot\right)$ and different feasible sets for the dual functions. Specifically, maximum mean discrepancy (MMD) (Smola et al., 2006) requires $\|f\|_{\mathcal{H}_k} \leq 1$ where $\mathcal{H}_k$ denotes the RKHS with kernel $k$; the Dudley metric (Dudley, 2002) requires $\|f\|_{BL} \leq 1$ where $\|f\|_{BL} := \|f\|_\infty + \|\nabla f\|_2$; and Wasserstein distance (Arjovsky et al., 2017) requires $\|\nabla f\|_2 \leq 1$. These additional requirements on the dual function might incur some extra difficulty in practice. For example, with Wasserstein distance and the Dudley metric, we might need to include an extra gradient penalty (Gulrajani et al., 2017), which requires additional computation to take the gradient through a gradient. Meanwhile, the consistency of the surrogate loss under regularization is not clear. For MMD, we can obtain a closed-form solution for the dual function, which saves the cost of the inner optimization (Gretton et al., 2012), but with the tradeoff of requiring *two independent* samples in each outer optimization update. Moreover, MMD relies on the condition that the dual function lies in some RKHS, which introduces additional kernel parameters to be tuned and in practice may not be sufficiently flexible compared to neural networks.

### 3.4 A PRACTICAL ALGORITHM

We have derived a consistent stationary distribution correction estimator in the form of a $\min$-$\max$ saddle point optimization Equation 13. Here, we present a practical instantiation of GenDICE with a concrete objective and parametrization.

We choose the $\chi^2$-divergence, which is an $f$-divergence with $\phi\left(x\right) = \left(x-1\right)^2$ and $\phi^*\left(y\right) = y + \frac{y^2}{4}$. The objective becomes

$$J_{\chi^2}\left(\tau,u,f\right) = \left(1-\gamma\right) \mathbb{E}_{\mu_0\pi}\left[f\left(s,a\right)\right] + \gamma\mathbb{E}_{\mathcal{T}_p}\left[\tau\left(s,a\right) f\left(s',a'\right)\right]$$

$$- \mathbb{E}_p\left[\tau\left(s,a\right) \left(f\left(s,a\right) + \frac{1}{4}f^2\left(s,a\right)\right)\right] + \lambda\left(\mathbb{E}_p\left[u\tau\left(s,a\right) - u\right] - \frac{u^2}{2}\right). \quad (14)$$

There two major reasons for adopting $\chi^2$-divergence:

**i)** In the behavior-agnostic OPE problem, we mainly use the ratio correction function for estimating $\widehat{\mathbb{E}}_p\left[\hat{\tau}\left(s,a\right) R\left(s,a\right)\right]$, which is an expectation. Recall that the error between the estimate and ground-truth can then be bounded by total variation, which is a lower bound of $\chi^2$-divergence.

**ii)** For the alternative divergences, the conjugate of the $KL$-divergence involves $\exp\left(\cdot\right)$, which may lead to instability in optimization; while the IPM variants introduce extra constraints on dual function, which may be difficult to be optimized. The conjugate function of $\chi^2$-divergence enjoys suitable numerical properties and provides squared regularization. We have provided an empirical ablation study that investigates the alternative divergences in Section 6.3.

To parameterize the correction ratio $\tau$ and dual function $f$ we use neural networks, $\tau\left(s, a\right) = \mathrm{nn}_{w_\tau}\left(s, a\right)$ and $f\left(s, a\right) = \mathrm{nn}_{w_f}\left(s, a\right)$, where $w_\tau$ and $w_f$ denotes the parameters of $\tau$ and $f$ respectively. Since the optimization requires $\tau$ to be non-negative, we add an extra positive neuron, such as $\exp\left(\cdot\right)$, $\log\left(1 + \exp\left(\cdot\right)\right)$ or $\left(\cdot\right)^2$ at the final layer of $\mathrm{nn}_{w_\tau}\left(s, a\right)$. We empirically compare the different positive neurons in Section 6.3.

For these representations, and unbiased gradient estimator $\nabla_{(w_\tau, u, w_f)} J\left(\tau, u, f\right)$ can be obtained straightforwardly, as shown in Appendix B. This allows us to apply stochastic gradient descent to solve the saddle-point problem Equation 14 in a scalable manner, as illustrated in Algorithm 1.

## 4 THEORETICAL ANALYSIS

We provide a theoretical analysis for the proposed GenDICE algorithm, following a similar learning setting and assumptions to (Nachum et al., 2019).

**Assumption 2** *The target stationary correction are bounded, $\|\tau^*\|_\infty \leq C < \infty$.*

The main result is summarized in the following theorem. A formal statement, together with the proof, is given in Appendix C.

**Theorem 3 (Informal)** *Under mild conditions, with learnable $\mathcal{F}$ and $\mathcal{H}$, the error in the objective between the GenDICE estimate, $\hat{\tau}$, to the solution $\tau^*\left(s, a\right) = \frac{u(s,a)}{p(s,a)}$ is bounded by*

$$\mathbb{E}\left[J\left(\hat{\tau}\right) - J\left(\tau^*\right)\right] = \widetilde{\mathcal{O}}\left(\epsilon_{approx}\left(\mathcal{F}, \mathcal{H}\right) + \frac{1}{\sqrt{N}} + \epsilon_{opt}\right),$$

*where $\mathbb{E}\left[\cdot\right]$ is w.r.t. the randomness in $\mathcal{D}$ and in the optimization algorithms, $\epsilon_{opt}$ is the optimization error, and $\epsilon_{approx}\left(\mathcal{F}, \mathcal{H}\right)$ is the approximation induced by $\left(\mathcal{F}, \mathcal{H}\right)$ for parametrization of $\left(\tau, f\right)$.*

The theorem shows that the suboptimality of GenDICE's solution, measured in terms of the objective function value, can be decomposed into three terms: (1) the approximation error $\epsilon_{approx}$, which is controlled by the representation flexibility of function classes; (2) the estimation error due to sample randomness, which decays at the order of $1/\sqrt{N}$; and (3) the optimization error, which arises from the suboptimality of the solution found by the optimization algorithm. As discussed in Appendix C, in special cases, this suboptimality can be bounded below by a divergence between $\hat{\tau}$ and $\tau^*$, and therefore directly bounds the error in the estimated policy value.

There is also a tradeoff between these three error terms. With more flexible function classes (*e.g.*, neural networks) for $\mathcal{F}$ and $\mathcal{H}$, the approximation error $\epsilon_{approx}$ becomes smaller. However, it may increase the estimation error (through the constant in front of $1/\sqrt{N}$) and the optimization error (by solving a harder optimization problem). On the other hand, if $\mathcal{F}$ and $\mathcal{H}$ are linearly parameterized, estimation and optimization errors tend to be smaller and can often be upper-bounded explicitly in Appendix C.3. However, the corresponding approximation error will be larger.

## 5 RELATED WORK

**Off-policy Policy Evaluation** Off-policy policy evaluation with importance sampling (IS) has has been explored in the contextual bandits (Strehl et al., 2010; Dudík et al., 2011; Wang et al., 2017), and episodic RL settings (Murphy et al., 2001; Precup et al., 2001), achieving many empirical successes (e.g., Strehl et al., 2010; Dudík et al., 2011; Bottou et al., 2013). Unfortunately, IS-based methods suffer from exponential variance in long-horizon problems, known as the "curse of horizon" (Liu et al., 2018). A few variance-reduction techniques have been introduced, but still cannot eliminate this fundamental issue (Jiang & Li, 2016; Thomas & Brunskill, 2016; Guo et al., 2017). By rewriting the accumulated reward as an expectation w.r.t. a stationary distribution, Liu et al. (2018); Gelada & Bellemare (2019) recast OPE as estimating a correction ratio function, which significantly alleviates variance. However, these methods still require the off-policy data to be collected by a *single and known* behavior policy, which restricts their practical applicability. The only published algorithm in the literature, to the best of our knowledge, that solves agnostic-behavior off-policy evaluation is DualDICE (Nachum et al., 2019). However, DualDICE was developed for

discounted problems and its results become unstable when the discount factor approaches 1 (see below). By contrast, GenDICE can cope with the more challenging problem of undiscounted reward estimation in the general behavior-agnostic setting.

Note that standard model-based methods (Sutton & Barto, 1998), which estimate the transition and reward models directly then calculate the expected reward based on the learned model, are also applicable to the behavior-agnostic setting considered here. Unfortunately, model-based methods typically rely heavily on modeling assumptions about rewards and transition dynamics. In practice, these assumptions do not always hold, and the evaluation results can become unreliable.

**Markov Chain Monte Carlo**    Classical MCMC (Brooks et al., 2011; Gelman et al., 2013) aims at sampling from $\mu^\pi$ by iteratively simuliting from the transition operator. It requires continuous interaction with the transition operator and heavy computational cost to update many particles. Amortized SVGD (Wang & Liu, 2016) and Adversarial MCMC (Song et al., 2017; Li et al., 2019) alleviate this issue via combining with neural network, but they still interact with the transition operator directly, *i.e.*, in an on-policy setting. The major difference of our GenDICE is the learning setting: we only access the off-policy dataset, and cannot sample from the transition operator. The proposed GenDICE leverages stationary density ratio estimation for approximating the stationary quantities, which distinct it from classical methods.

**Density Ratio Estimation**    Density ratio estimation is a fundamental tool in machine learning and much related work exists. Classical density ratio estimation includes moment matching (Gretton et al., 2008), probabilistic classification (Bickel et al., 2007), and ratio matching (Nguyen et al., 2008; Sugiyama et al., 2008; Kanamori et al., 2009). These classical methods focus on estimating the ratio between two distributions with samples from both of them, while GenDICE estimates the density ratio to a stationary distribution of a transition operator, from which even one sample is difficult to obtain.

**PageRank**    Yao & Schuurmans (2013) developed a reverse-time RL framework for PageRank via solving a reverse Bellman equation, which is less sensitive to graph topology and shows faster adaptation with graph change. However, Yao & Schuurmans (2013) still considers the online manner, which is different with our OPR setting.

# 6    EXPERIMENTS

In this section, we evaluate GenDICE on OPE and OPR problems. For OPE, we use one or multiple behavior policies to collect a fixed number of trajectories at some fixed trajectory length. This data is used to recover a correction ratio function for a target policy $\pi$ that is then used to estimate the average reward in two different settings: *i*) average reward; and *ii*) discounted reward. In both settings, we compare with a model-based approach and step-wise weighted IS (Precup et al., 2000). We also compare to Liu et al. (2018) (referred to as "IPS" here) in the Taxi domain with a learned behavior policy[1]. We specifically compare to DualDICE (Nachum et al., 2019) in the discounted reward setting, which is a direct and current state-of-the-art baseline. For OPR, the main comparison is with the model-based method, where the transition operator is empirically estimated and stationary distribution recovered via an exact solver. We validate GenDICE in both tabular and continuous cases, and perform an ablation study to further demonstrate its effectiveness. All results are based on 20 random seeds, with mean and standard deviation plotted. Our code is publicly available at `https://github.com/zhangry868/GenDICE`.

## 6.1    TABULAR CASE

**Offline PageRank on Graphs**    One direct application of GenDICE is off-line PageRank (OPR). We test GenDICE on a Barabasi-Albert (BA) graph (synthetic), and two real-world graphs, Cora and Citeseer. Details of the graphs are given in Appendix D. We use the $\log KL$-divergence between estimated stationary distribution and the ground truth as the evaluation metric, with the ground truth computed by an exact solver based on the exact transition operator of the graphs. We compared GenDICE with model-based methods in terms of the sample efficiency. From the results in Figure 1, GenDICE outperforms the model-based method when limited data is given. Even with $20k$ samples for a BA graph with 100 nodes, where a transition matrix has $10k$ entries, GenDICE still

---

[1]We used the released implementation of IPS (Liu et al., 2018) from `https://github.com/zt95/infinite-horizon-off-policy-estimation`.

shows better performance in the offline setting. This is reasonable since GenDICE directly estimates the stationary distribution vector or ratio, while the model-based method needs to learn an entire transition matrix that has many more parameters.

**Off-Policy Evaluation with Taxi**   We use a similar taxi domain as in Liu et al. (2018), where a grid size of $5 \times 5$ yields 2000 states in total ($25 \times 16 \times 5$, corresponding to 25 taxi locations, 16 passenger appearance status and 5 taxi status). We set the target policy to a final policy $\pi$ after running tabular Q-learning for 1000 iterations, and set another policy $\pi_+$ after 950 iterations as the base policy. The behavior policy is a mixture controlled by $\alpha$ as $\pi_b = (1 - \alpha)\pi + \alpha\pi_+$. For the model-based method, we use a tabular representation for the reward and transition functions, whose entries are estimated from behavior data. For IS and IPS, we fit a policy via behavior cloning to estimate the policy ratio. In this specific setting, our methods achieve better results compared to IS, IPS and the model-based method. Interestingly, with longer horizons, IS cannot improve as much as other methods even with more data, while GenDICE consistently improve and achieves much better results than the baselines. DualDICE

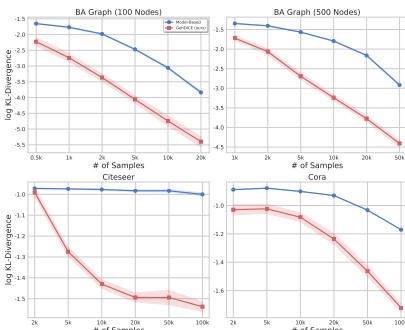

Figure 1: Stationary Distribution Estimation on BA and real-world graphs. Each plot shows the $\log KL$-divergence of GenDICE and model-based method towards the number of samples.

only works with $\gamma < 1$. GenDICE is more stable than DualDICE when $\gamma$ becomes larger (close to 1), while still showing competitive performance for smaller discount factors $\gamma$.

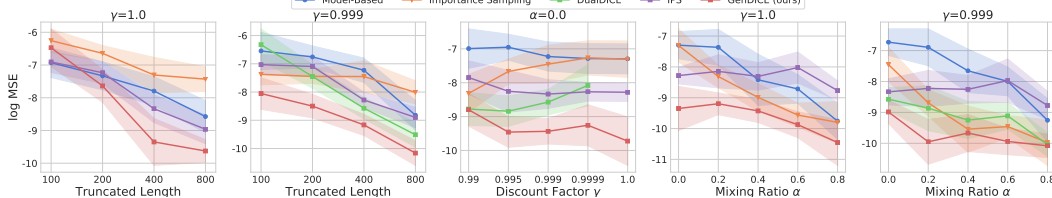

Figure 2: Results on Taxi Domain. The plots show log MSE of the tabular estimator across different trajectory lengths, different discount factors and different behavior policies ($x$-axis).

## 6.2   CONTINUOUS CASE

We further test our method for OPE on three control tasks: a discrete-control task Cartpole and two continuous-control tasks Reacher and HalfCheetah. In these tasks, observations (or states) are continuous, thus we use neural network function approximators and stochastic optimization. Since DualDICE (Nachum et al., 2019) has shown the state-of-the-art performance on discounted OPE, we mainly compare with it in the discounted reward case. We also compare to IS with a learned policy via behavior cloning and a neural model-based method, similar to the tabular case, but with neural network as the function approximator. All neural networks are feed-forward with two hidden layers of dimension 64 and tanh activations. More details can be found in Appendix D.

Due to limited space, we put the discrete control results in Appendix E and focus on the more challenging continuous control tasks. Here, the good performance of IS and model-based methods in Section 6.1 quickly deteriorates as the environment becomes complex, *i.e.*, with a continuous action space. Note that GenDICE is able to maintain good performance in this scenario, even when using function approximation and stochastic optimization. This is reasonable because of the difficulty of fitting to the coupled policy-environment dynamics with a continuous action space. Here we also *empirically* validate GenDICE with off-policy data collected by multiple policies.

As illustrated in Figure 3, all methods perform better with longer trajectory length or more trajectories. When $\alpha$ becomes larger, *i.e.*, the behavior policies are closer to the target policy, all methods performs better, as expected. Here, GenDICE demonstrates good performance both on average-reward and discounted reward cases in different settings. The right two figures in each row show the $\log$ MSE curve versus optimization steps, where GenDICE achieves the smallest loss. In the discounted reward case, GenDICE shows significantly better and more stable performance than the

strong baseline, DualDICE. Figure 4 also shows better performance of GenDICE than all baselines in the more challenging HalfCheetah domain.

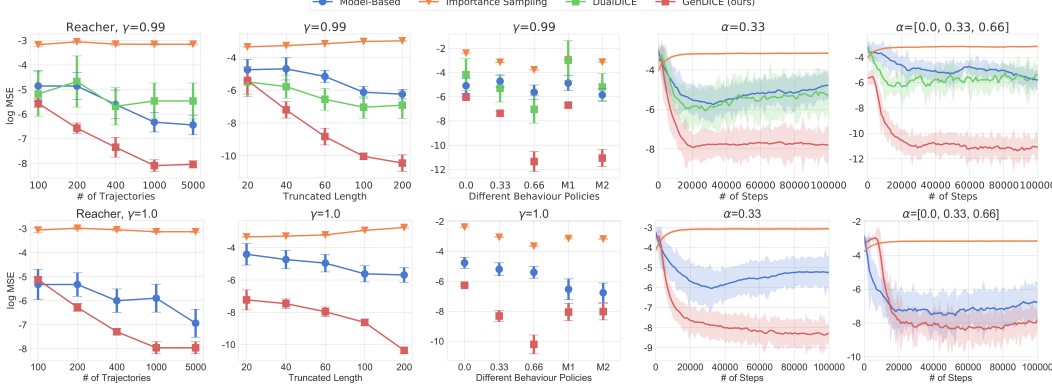

Figure 3: Results on Reacher. The left three plots in the first row show the $\log$ MSE of estimated average per-step reward over different numbers of trajectories, truncated lengths, and behavior policies (M1 and M2 mean off-policy set collected by multiple behavior policies with $\alpha = [0.0, 0.33]$ and $\alpha = [0.0, 0.33, 0.66]$). The right two figures show the loss curves towards the optimization steps. Each plot in the second row shows the average reward case.

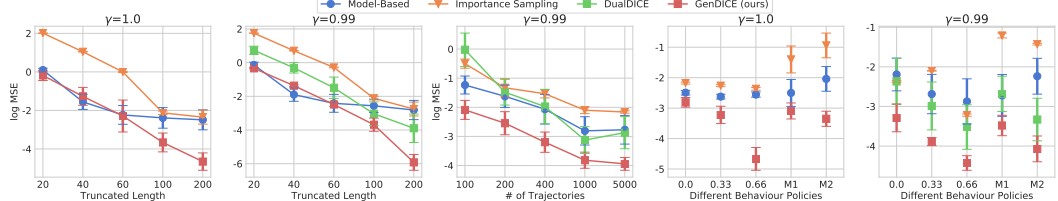

Figure 4: Results on HalfCheetah. Plots from left to the right show the $\log$ MSE of estimated average per-step reward over different truncated lengths, numbers of trajectories, and behavior policies in discounted and average reward cases.

### 6.3 ABLATION STUDY

Finally, we conduct an ablation study on GenDICE to study its robustness and implementation sensitivities. We investigate the effects of learning rate, activation function, discount factor, and the specifically designed ratio constraint. We further demonstrate the effect of the choice of divergences and the penalty weight.

**Effects of the Learning Rate**  Since we are using neural network as the function approximator, and stochastic optimization, it is necessary to show sensitivity to the learning rate with $\{0.0001, 0.0003, 0.001, 0.003\}$, with results in Figure 5. When $\alpha = 0.33$, *i.e.*, the OPE tasks are relatively easier and GenDICE obtains better results at all learning rate settings. However, when $\alpha = 0.0$, *i.e.*, the estimation becomes more difficult and only GenDICE only obtains reasonable results with the larger learning rate. Generally, this ablation study shows that the proposed method is not sensitive to the learning rate, and is easy to train.

**Activation Function of Ratio Estimator**  We further investigate the effects of the activation function on the last layer, which ensure the non-negative outputs required for the ratio. To better understand which activation function will lead to stable trainig for the neural correction estimator, we empirically compare using *i)* $(\cdot)^2$; *ii)* $\log(1 + \exp(\cdot))$; and *iii)* $\exp(\cdot)$. In practice, we use the $(\cdot)^2$ since it achieves low variance and better performance in most cases, as shown in Figure 5.

**Effects of Discount Factors**  We vary $\gamma \in \{0.95, 0.99, 0.995, 0.999, 1.0\}$ to probe the sensitivity of GenDICE. Specifically, we compare to DualDICE, and find that GenDICE is stable, while DualDICE becomes unstable when the $\gamma$ becomes large, as shown in Figure 6. GenDICE is also more general than DualDICE, as it can be applied to both the average and discounted reward cases.

**Effects of Ratio Constraint**  In Section 3, we highlighted the importance of the ratio constraint. Here we investigate the trivial solution issue without the constraint. The results in Figure 6 demonstrate the necessity of adding the constraint penalty, since a trivial solution prevents an accurate corrector from being recovered (green line in left two figures).

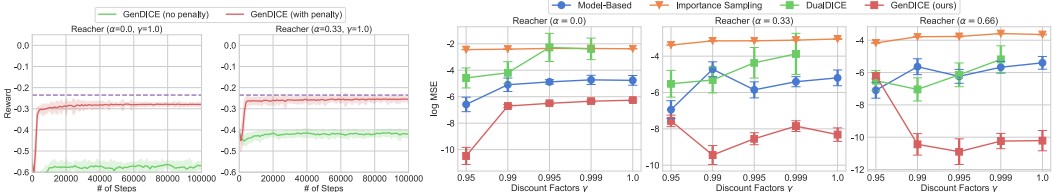

Figure 5: Results of ablation study with different learning rates and activation functions. The plots show the log MSE of estimated average per-step reward over training and different behavior policies.

Figure 6: Results of ablation study with constraint penalty and discount factors. The left two figures show the effect of ratio constraint on estimating average per-step reward. The right three figures show the log MSE for average per-step reward over training and different discount factor $\gamma$.

**Effects of the Choice of Divergences** We empirically test the GenDICE with several other alternative divergences, *e.g.*, Wasserstein-1 distance, Jensen-Shannon divergence, $KL$-divergence, Hellinger divergence, and MMD. To avoid the effects of other factors in the estimator, *e.g.*, function parametrization, we focus on the offline PageRank task on BA graph with 100 nodes and $10k$ offline samples. All the experiments are evaluated with 20 random trials. To ensure the dual function to be 1-Lipchitz, we add the gradient penalty. Besides, we use a learned Gaussian kernel in MMD, similar to Li et al. (2017). As we can see in Figure 7(a), the GenDICE estimator is compatible with many different divergences. Most of the divergences, with appropriate extra techniques to handle the difficulties in optimization and carefully

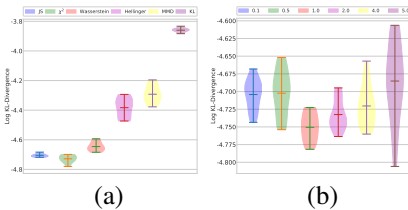

(a)          (b)

Figure 7: Results of ablation study with (a) different divergence and (b) weight of penalty $\lambda$. The plots show the log $KL$-Divergence of OPR on Barabasi-Albert graph.

tuning for extra parameters, can achieve similar performances, consistent with phenomena in the variants of GANs (Lucic et al., 2018). However, $KL$-divergence is an outlier, performing noticeably worse, which might be caused by the ill-behaved $\exp(\cdot)$ in its conjugate function. The $\chi^2$-divergence and JS-divergence are better, which achieve good performances with fewer parameters to be tuned.

**Effects of the Penalty Weight** The results of different penalty weights $\lambda$ are illustrated in Figure 7(b). We vary the $\lambda \in [0.1, 5]$ with $\chi^2$-divergence. Within a large range of $\lambda$, the performances of the proposed GenDICE are quite consistent, which justifies Theorem 1. The penalty multiplies with $\lambda$. Therefore, with $\lambda$ increases, the variance of the stochastic gradient estimator also increases, which explains the variance increasing in large $\lambda$ in Figure 7(b). In practice, $\lambda = 1$ is a reasonable choice for general cases.

## 7 CONCLUSION

In this paper, we proposed a novel algorithm GenDICE for general stationary distribution correction estimation, which can handle both the discounted and average stationary distribution given multiple behavior-agnostic samples. Empirical results on off-policy evaluation and offline PageRank show the superiority of proposed method over the existing state-of-the-art methods.

### ACKNOWLEDGMENTS

The authors would like to thank Ofir Nachum, the rest of the Google Brain team and the anonymous reviewers for helpful discussions and feedback.

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

# Appendix

## A PROPERTIES OF GENDICE

For notation simplicity, we denote $x = (s, a) \in \Omega := S \times A$ and $\mathbf{P}^\pi (x'|x) := \pi (a'|s') \mathbf{P} (s'|s, a)$. Also define $\|f\|_{p,2} := \langle f, f \rangle_p = \int f (x)^2 p (x) dx$. We make the following assumption to ensure the existence of the stationary distribution. Our discussion is all based on this assumption.

**Assumption 1** *Under the target policy, the resulted state-action transition operator $\mathcal{T}$ has a unique stationary distribution in terms of the divergence $D (\cdot||\cdot)$.*

If the total variation divergence is selected, the Assumption 1 requires the transition operator should be ergodic, as discussed in Meyn & Tweedie (2012).

### A.1 CONSISTENCY OF THE ESTIMATOR

**Theorem 1** *For arbitrary $\lambda > 0$, the solution to the optimization Eqn (11) is $\frac{u(s,a)}{p(s,a)}$ for $\gamma \in (0, 1]$.*

**Proof** For $\gamma \in (0, 1)$, there is not degenerate solutions to $D \left( \left( \mathcal{T}^p_{\gamma,\mu_0} \circ \tau \right) ||p \cdot \tau \right)$. The optimal solution is a density ratio. Therefore, the extra penalty $\left( \mathbb{E}_{p(x)} [\tau (x)] - 1 \right)^2$ does not affect the optimality for $\forall \lambda > 0$.

When $\gamma = 1$, for $\forall \lambda > 0$, recall both $D \left( \left( \mathcal{T}^p_{\gamma,\mu_0} \circ \tau \right) ||p \cdot \tau \right)$ and $\left( \mathbb{E}_{p(x)} [\tau (x)] - 1 \right)^2$ are non-negative, and the the density ratio $\frac{\mu(x)}{p(x)}$ leads to zero for both terms. Then, the density ratio is a solution to $J (\tau)$. For any other non-negative function $\tau (x) \geq 0$, if it is the optimal solution to $J (\tau)$, then, we have

$$D \left( \left( \mathcal{T}^p_{\gamma,\mu_0} \circ \tau \right) ||p \cdot \tau \right) = 0 \quad \Rightarrow \quad p (x') \tau (x') = \left( \mathcal{T}^p_{\gamma,\mu_0} \circ \tau \right) (x') = \int \mathbf{P}^\pi (x'|x) \tau (x) dx, \quad (15)$$

$$\left( \mathbb{E}_{p(x)} [\tau (x)] - 1 \right)^2 = 0 \quad \Rightarrow \quad \mathbb{E}_{p(x)} [\tau (x)] = 1. \quad (16)$$

We denote $\mu (x) = p (x) \tau (x)$, which is clearly a density function. Then, the optimal conditions in Equation 15 imply

$$\mu (x') = \int \mathbf{P}^\pi (x'|x) \mu (x) dx,$$

or equivalently, $\mu$ is the stationary distribution of $\mathcal{T}$. We have thus shown the optimal $\tau (x) = \frac{\mu(x)}{p(x)}$ is the target density ratio. ∎

### A.2 CONVEXITY OF THE OBJECTIVE

**Proof** Since the $\phi$ is convex, we consider the Fenchel dual representation of the $f$-divergence $D_\phi \left( \left( \mathcal{T}^p_{\gamma,\mu_0} \circ \tau \right) ||p \cdot \tau \right)$, *i.e.*,

$$D_\phi \left( \left( \mathcal{T}^p_{\gamma,\mu_0} \circ \tau \right) ||p \cdot \tau \right) = \max_{f \in \Omega \to \mathbb{R}} \ell (\tau, f)$$

$$:= (1 - \gamma) \mathbb{E}_{\mu_0 \pi} [f (x)] + \gamma \mathbb{E}_{\mathcal{T}_p(x,x')} [\tau (x) f (x')] - \mathbb{E}_{p(x)} [\tau (x) \phi^* (f (x))]. \quad (17)$$

It is obviously $\ell (\tau, f)$ is convex in $\tau$ for each $f$, then, $D_\phi \left( \left( \mathcal{T}^p_{\gamma,\mu_0} \circ \tau \right) ||p \cdot \tau \right)$ is convex. The term $\lambda \left( \mathbb{E}_p (\tau) - 1 \right)^2$ is also convex, which concludes the proof. ∎

---

**Algorithm 1** GenDICE (with function approximators)

**Inputs**: Convex function $\phi$ and its Fenchel conjugate $\phi^*$, off-policy data $\mathcal{D} = \{(s^{(i)}, a^{(i)}, r^{(i)}, s'^{(i)})\}_{i=1}^N$, initial state $s_0 \sim \mu_0$, target policy $\pi$, distribution corrector $\text{nn}_{w_\tau}(\cdot, \cdot), \text{nn}_{w_f}(\cdot, \cdot)$, constraint scalar $u$, learning rates $\eta_\tau, \eta_f, \eta_u$, number of iterations $K$, batch size $B$.

**for** $t = 1, \ldots, K$ **do**

    Sample batch $\{(s^{(i)}, a^{(i)}, r^{(i)}, s'^{(i)})\}_{i=1}^B$ from $\mathcal{D}$.

    Sample batch $\{s_0^{(i)}\}_{i=1}^B$ from $\mu_0$.

    Sample actions $a'^{(i)} \sim \pi(s'^{(i)})$, for $i = 1, \ldots, B$.

    Sample actions $a_0^{(i)} \sim \pi(s_0^{(i)})$, for $i = 1, \ldots, B$.

    Compute empirical loss $\hat{J}_{\chi^2}(\tau, u, f) = (1 - \gamma) \mathbb{E}_{\mu_0 \pi}[f(s, a)] + \gamma \mathbb{E}_{\mathcal{T}_p}[\tau(s, a) f(s', a')]$

    $-\mathbb{E}_p\left[\tau(s, a)\left(f(s, a) + \frac{1}{4}f^2(s, a)\right)\right] + \lambda\left(\mathbb{E}_p[u\tau(s, a) - u] - \frac{u^2}{2}\right).$

    Update $w_\tau \leftarrow w_\tau - \eta_\tau \nabla_{\theta_\tau} \hat{J}_{\chi^2}.$

    Update $w_f \leftarrow w_f + \eta_f \nabla_{\theta_f} \hat{J}_{\chi^2}.$

    Update $u \leftarrow u + \eta_u \nabla_u \hat{J}_{\chi^2}.$

**end for**

**Return** $\text{nn}_{w_\tau}.$

---

## B  ALGORITHM DETAILS

We provide the unbiased gradient estimator for $\nabla_{w_\tau, u, w_f} J(\tau, u, f)$ in Eqn (14) below:

$$\nabla_{w_\tau} J_{\chi^2}(\tau, u, f) = \gamma \mathbb{E}_{\mathcal{T}_p}[\nabla_{w_\tau} \tau(s, a) f(s', a')] - \mathbb{E}_p\left[\nabla_{w_\tau} \tau(s, a)\left(f(s, a) + \frac{1}{4}f^2(s, a)\right)\right]$$

$$+\lambda u \mathbb{E}_p[\nabla_{w_\tau} \tau(s, a)], \tag{18}$$

$$\nabla_u J_{\chi^2}(\tau, u, f) = \lambda\left(\mathbb{E}_p[\tau(s, a) - 1] - u\right), \tag{19}$$

$$\nabla_{w_f} J_{\chi^2}(\tau, u, f) = (1 - \gamma)\mathbb{E}_{\mu_0 \pi}[\nabla_{w_f} f(s, a)] + \gamma \mathbb{E}_{\mathcal{T}_p}[\tau(s, a)\nabla_{w_f} f(s', a')] \tag{20}$$

$$-\mathbb{E}_p\left[\tau(s, a)\left(1 + \frac{1}{2}f(s, a)\right)\nabla_{w_f} f(s, a)\right].$$

Then, we have the psuedo code which applies SGD for solving Eqn (14).

## C  PROOF OF THEOREM 3

For convenience, we repeat here the notation defined in the main text. The saddle-point reformulation of the objective function of GenDICE is:

$$J(\tau, u, f) := (1 - \gamma)\mathbb{E}_{\mu_0 \pi}[f(x')] + \gamma \mathbb{E}_{\mathcal{T}_p(x, x')}[\tau(x) f(x')]$$

$$- \mathbb{E}_{p(x)}[\tau(x)\phi^*(f(x))] + \lambda\left(\mathbb{E}_{p(x)}[u\tau(x) - u] - \frac{1}{2}u^2\right).$$

To avoid the numerical infinity in $D_\phi(\cdot||\cdot)$, we induced the bounded version as

$$J(\tau) := \max_{\|f\|_\infty \leq C, u} J(\tau, u, f) = D_\phi^C\left((\mathcal{T}_{\gamma, \mu_0}^p \circ \tau)||p \cdot \tau\right) + \frac{\lambda}{2}\left(\mathbb{E}_{p(x)}[\tau(x)] - 1\right)^2,$$

in which $D_\phi^C(\cdot||\cdot)$ is still a valid divergence, and therefore the optimal solution $\tau^*$ is still the stationary density ratio $\frac{\mu(x)}{p(x)}$. We denote the $\hat{J}(\tau, \mu, f)$ as the empirical surrogate of $J(\tau, \mu, f)$ on samples $\mathcal{D} = \left((x, x')_{i=1}^N\right) \sim \mathcal{T}_{\gamma, \mu_0}^p(x, x')$ with the optimal solution in $(\mathcal{H}, \mathcal{F}, \mathbb{R})$ as $\left(\hat{\tau}_{\mathcal{H}}^*, \hat{u}^*, \hat{f}_{\mathcal{F}}^*\right)$. Furthermore, denote

$$\tau_{\mathcal{H}}^* = \arg\min_{\tau \in \mathcal{H}} J(\tau),$$

$$\tau^* = \arg\min_{\tau \in S \times A \to \mathbb{R}} J(\tau)$$

with optimal $(f^*, u^*)$, and

$$L\left(\tau\right) = \max_{f \in \mathcal{F}, u \in \mathbb{R}} J\left(\tau, u, f\right),$$

$$\widehat{L}\left(\tau\right) = \max_{f \in \mathcal{F}, u \in \mathbb{R}} \widehat{J}\left(\tau, u, f\right).$$

We apply some optimization algorithm for $\widehat{J}\left(\tau, u, f\right)$ over space $(\mathcal{H}, \mathcal{F}, \mathbb{R})$, leading to the output $\left(\hat{\tau}, \widehat{u}, \widehat{f}\right)$.

Under Assumption 2, we need only consider $\|\tau\|_\infty \leq C$, then, the corresponding dual $u = \mathbb{E}_p\left(\tau\right) - 1 \Rightarrow u \in U := \{|u| \leq (C+1)\}$. We choose the $\phi^*\left(\cdot\right)$ is a $\kappa$-Lipschitz continuous, then, the $J\left(\tau, u, f\right)$ is a $C_{\mathbf{P}^\pi, \kappa, \lambda} = \max\left\{\left(\gamma \|\mathbf{P}^\pi\|_{p,\infty} + (1-\gamma)\left\|\frac{\mu_0 \pi}{p}\right\|_{p,\infty} + \kappa\right) C, (C+1)\left(\lambda + \frac{1}{2}\right)\right\}$-Lipschitz continuous function w.r.t. $(f, u)$ with the norm $\|(f, u)\|_{p,1} := \int |f\left(x\right)| p\left(x\right) dx + |u|$, and $C_{\phi, C, \lambda} := \left(C + \lambda\left(C+1\right) + \max_{t \in \{-C, C\}}\left(-\phi\left(t\right)\right)\right)$-Lipschitz continuous function w.r.t. $\tau$ with the norm $\|\tau\|_{p,1} := \int |\tau\left(x\right)| p\left(x\right) dx$.

We consider the error between $\hat{\tau}$ and $\tau^*$ using standard arguments (Shalev-Shwartz & Ben-David, 2014; Bach, 2014), *i.e.*,

$$d\left(\hat{\tau}, \tau^*\right) := J\left(\hat{\tau}\right) - J\left(\tau^*\right).$$

The discrepancy $d\left(\tau, \tau^*\right) \geq 0$ and $d\left(\tau, \tau^*\right) = 0$ if and only if $p \cdot \tau$ is stationary distribution of $\mathcal{T}$ in the weak sense of $D_\phi\left(\cdot \| \cdot\right)$.

**Remark:** In some special cases, the suboptimality also implies the distance between $\hat{\tau}$ and $\tau^*$. Specifically, for $\gamma = 1$, if the transition operator $\mathbf{P}^\pi$ can be represented as $\mathbf{P}^\pi = \mathcal{Q}\Lambda\mathcal{Q}^{-1}$ where $\mathcal{Q}$ denotes the (countable) eigenfunctions and $\Lambda$ denotes the diagonal matrix with eigenvalues, the largest of which is 1. We consider $\phi\left(\cdot\right)$ as identity and $f \in \mathcal{F} := \left\{\text{span}\left(\mathcal{Q}\right), \|f\|_{p,2} \leq 1\right\}$, then the $d\left(\tau, \tau^*\right)$ will bounded from below by a metric between $\tau$ and $\tau^*$. Particularly, we have

$$D_\phi\left(\left(\mathcal{T}_{\gamma, \mu_0}^p \circ \tau\right) \| p \cdot \tau\right) = \max_{f \in \mathcal{F}} \mathbb{E}_{\mathcal{T}_p(x, x')}\left[\tau\left(x\right) f\left(x'\right)\right] - \mathbb{E}_{p(x)}\left[\tau\left(x\right) f\left(x\right)\right] = \|\tau - \mathbf{P}^\pi \circ \tau\|_{p,2}.$$

Rewrite $\tau = \alpha\tau^* + \zeta$, where $\zeta \in \text{span}\left(\mathcal{Q}_{\setminus \tau^*}\right)$, then

$$D_\phi\left(\left(\mathcal{T}_{\gamma, \mu_0}^p \circ \tau\right) \| p \cdot \tau\right) = \|\alpha\tau^* - \alpha\mathbf{P}^\pi \circ \tau^* + \zeta - \mathbf{P}^\pi \circ \zeta\|_{p,2} = \|\zeta - \mathbf{P}^\pi \circ \zeta\|_{p,2}.$$

Recall the optimality of $\tau^*$, *i.e.*, $D_\phi\left(\left(\mathcal{T}_{\gamma, \mu_0}^p \circ \tau\right)^* \| p \cdot \tau^*\right) = 0$, we have

$$d\left(\tau, \tau^*\right) = J\left(\tau\right) \geq \|\zeta - \mathbf{P}^\pi \circ \zeta\|_{p,2} := \|(\tau - \tau^*)\|_{p,2,(\mathbf{P}^\pi - I)}.$$

## C.1 Error Decomposition

We start with the following error decomposition:

$$d\left(\hat{\tau}, \tau^*\right) := J\left(\hat{\tau}\right) - J\left(\tau^*\right) = \underbrace{J\left(\hat{\tau}\right) - J\left(\hat{\tau}_\mathcal{H}^*\right)}_{\epsilon_1} + \underbrace{J\left(\hat{\tau}_\mathcal{H}^*\right) - J\left(\tau^*\right)}_{\epsilon_2}.$$

- For $\epsilon_1$, we have

$$\epsilon_1 = J\left(\hat{\tau}\right) - L\left(\hat{\tau}\right) + L\left(\hat{\tau}\right) - L\left(\hat{\tau}_\mathcal{H}^*\right) + L\left(\hat{\tau}_\mathcal{H}^*\right) - J\left(\hat{\tau}_\mathcal{H}^*\right).$$

We consider the terms one-by-one. By definition, we have

$$\begin{aligned} J\left(\hat{\tau}\right) - L\left(\hat{\tau}\right) &= \max_{f, \mu} J\left(\hat{\tau}, u, f\right) - \max_{f \in \mathcal{F}, \mu} J\left(\hat{\tau}, u, f\right) \\ &\leq C_{\mathbf{P}^\pi, \kappa, \lambda} \underbrace{\sup_{f_1, u_1 \in U} \inf_{f_2 \in \mathcal{F}, u_2 \in U} \|(f_1, u_1) - (f_2, u_2)\|_{p,1}}_{\epsilon_{approx}(\mathcal{F})}, \end{aligned} \quad (21)$$

which is induced by introducing $\mathcal{F}$ for dual approximation.

For the third term $L\left(\hat{\tau}_\mathcal{H}^*\right) - J\left(\hat{\tau}_\mathcal{H}^*\right)$, we have

$$L\left(\hat{\tau}_\mathcal{H}^*\right) - J\left(\hat{\tau}_\mathcal{H}^*\right) = \max_{f \in \mathcal{F}, u \in U} J\left(\hat{\tau}_\mathcal{H}^*, u, f\right) - \max_{f, u \in U} J\left(\hat{\tau}_\mathcal{H}^*, u, f\right) \leq 0.$$

For the term $L\left(\hat{\tau}\right) - L\left(\hat{\tau}_{\mathcal{H}}^*\right)$,

$$
\begin{aligned}
L\left(\hat{\tau}\right) - L\left(\hat{\tau}_{\mathcal{H}}^*\right) &= L\left(\hat{\tau}\right) - \widehat{L}\left(\hat{\tau}\right) + \underbrace{\widehat{L}\left(\hat{\tau}\right) - \widehat{L}\left(\hat{\tau}_{\mathcal{H}}^*\right)}_{\hat{\epsilon}_{opt}} + \widehat{L}\left(\hat{\tau}_{\mathcal{H}}^*\right) - L\left(\hat{\tau}_{\mathcal{H}}^*\right) \\
&\leq 2\sup_{\tau\in\mathcal{H}}\left|L\left(\tau\right) - \widehat{L}\left(\tau\right)\right| + \hat{\epsilon}_{opt} \\
&\leq 2\sup_{\tau\in\mathcal{H}}\left|\max_{f\in\mathcal{F},u\in U}J\left(\tau,u,f\right) - \max_{f\in\mathcal{F},u\in U}\widehat{J}\left(\tau,u,f\right)\right| + \hat{\epsilon}_{opt} \\
&\leq 2\sup_{\tau\in\mathcal{H}}\sup_{f\in\mathcal{F},u\in U}\left|J\left(\tau,u,f\right) - \widehat{J}\left(\tau,u,f\right)\right| + \hat{\epsilon}_{opt} \\
&= 2\cdot\epsilon_{est} + \hat{\epsilon}_{opt},
\end{aligned}
\tag{22}
$$

where we define $\epsilon_{est} := \sup_{\tau\in\mathcal{H},f\in\mathcal{F},u\in U}\left|J\left(\tau,u,f\right) - \widehat{J}\left(\tau,u,f\right)\right|$.

Therefore, we can now bound $\epsilon_1$ as

$$
\epsilon_1 \leq C_{\mathcal{T},\kappa,\lambda}\epsilon_{approx}\left(\mathcal{F}\right) + 2\epsilon_{est} + \hat{\epsilon}_{opt}.
$$

- For $\epsilon_2$, we have

$$
\begin{aligned}
\epsilon_2 &= J\left(\hat{\tau}_{\mathcal{H}}^*\right) - J\left(\tau_{\mathcal{H}}^*\right) + J\left(\tau_{\mathcal{H}}^*\right) - J\left(\tau^*\right) \\
&= J\left(\hat{\tau}_{\mathcal{H}}^*\right) - L\left(\hat{\tau}_{\mathcal{H}}^*\right) + L\left(\hat{\tau}_{\mathcal{H}}^*\right) - L\left(\tau_{\mathcal{H}}^*\right) + L\left(\tau_{\mathcal{H}}^*\right) - J\left(\tau_{\mathcal{H}}^*\right) + J\left(\tau_{\mathcal{H}}^*\right) - J\left(\tau^*\right).
\end{aligned}
$$

We consider the terms from right to left. For the term $J\left(\tau_{\mathcal{H}}^*\right) - J\left(\tau^*\right)$, we have

$$
\begin{aligned}
J\left(\hat{\tau}_{\mathcal{H}}^*\right) - J\left(\tau^*\right) &= J\left(\hat{\tau}_{\mathcal{H}}^*, \widehat{u}^*, \widehat{f}_{\mathcal{F}}^*\right) - J\left(\tau^*, \widehat{u}^*, \widehat{f}_{\mathcal{F}}^*\right) + \underbrace{J\left(\tau^*, \widehat{u}^*, \widehat{f}_{\mathcal{F}}^*\right) - J\left(\tau^*, u^*, f^*\right)}_{\leq 0} \\
&= J\left(\hat{\tau}_{\mathcal{H}}^*, \widehat{u}^*, \widehat{f}_{\mathcal{F}}^*\right) - J\left(\tau^*, \widehat{u}^*, \widehat{f}_{\mathcal{F}}^*\right) \leq C_{\phi,C,\lambda}\underbrace{\sup_{\tau_1}\inf_{\tau_2\in\mathcal{H}}\left\|\tau_1 - \tau_2\right\|_{p,1}}_{\epsilon_{approx}\left(\mathcal{H}\right)},
\end{aligned}
$$

which is induced by restricting the function space to $\mathcal{H}$. The second term is nonpositive, due to the optimality of $\left(u^*, f^*\right)$. The final inequality comes from the fact that $J\left(\tau,u,f\right)$ is $C_{\phi,C,\lambda}$-Lipschitz w.r.t. $\tau$.

For the term $L\left(\hat{\tau}_{\mathcal{H}}^*\right) - J\left(\tau_{\mathcal{H}}^*\right)$, by definition

$$
L\left(\tau_{\mathcal{H}}^*\right) - J\left(\tau_{\mathcal{H}}^*\right) = \max_{f\in\mathcal{F},u\in U}J\left(\tau_{\mathcal{H}}^*,f,u\right) - \max_{f,u\in U}J\left(\tau_{\mathcal{H}}^*,f,u\right) \leq 0.
$$

For the term $L\left(\hat{\tau}_{\mathcal{H}}^*\right) - L\left(\tau_{\mathcal{H}}^*\right)$, we have

$$
\begin{aligned}
L\left(\hat{\tau}_{\mathcal{H}}^*\right) - L\left(\tau_{\mathcal{H}}^*\right) &= L\left(\hat{\tau}_{\mathcal{H}}^*\right) - \widehat{L}\left(\hat{\tau}_{\mathcal{H}}^*\right) + \underbrace{\widehat{L}\left(\hat{\tau}_{\mathcal{H}}^*\right) - \widehat{L}\left(\tau_{\mathcal{H}}^*\right)}_{\leq 0} + \widehat{L}\left(\tau_{\mathcal{H}}^*\right) - L\left(\tau_{\mathcal{H}}^*\right) \\
&= 2\sup_{\tau\in\mathcal{H}}\left|L\left(\tau\right) - \widehat{L}\left(\tau\right)\right| \\
&= 2\sup_{\tau\in\mathcal{H},f\in\mathcal{F},u\in U}\left|J\left(\tau,u,f\right) - \widehat{J}\left(\tau,u,f\right)\right| \\
&= 2\cdot\epsilon_{est}.
\end{aligned}
$$

where the second term is nonpositive, thanks to the optimality of $\hat{\tau}_{\mathcal{H}}^*$.

Finally, for the term $J\left(\hat{\tau}_{\mathcal{H}}^*\right) - J\left(\tau_{\mathcal{H}}^*\right)$, using the same argument in Equation 21, we have

$$
J\left(\hat{\tau}_{\mathcal{H}}^*\right) - J\left(\tau_{\mathcal{H}}^*\right) \leq C_{\mathbf{P}^\pi,\kappa,\lambda}\epsilon_{approx}\left(\mathcal{F}\right).
$$

Therefore, we can bound $\epsilon_2$ by

$$
\epsilon_2 \leq C_{\phi,C,\lambda}\epsilon_{approx}\left(\mathcal{H}\right) + C_{\mathbf{P}^\pi,\kappa,\lambda}\left(\mathcal{F}\right) + 2\epsilon_{est}.
$$

In sum, we have

$$
d\left(\hat{\tau},\tau^*\right) \leq 4\epsilon_{est} + \hat{\epsilon}_{opt} + 2C_{\mathbf{P}^\pi,\kappa,\lambda}\epsilon_{approx}\left(\mathcal{F}\right) + C_{\phi,C,\lambda}\epsilon_{approx}\left(\mathcal{H}\right).
$$

In the following sections, we will bound the $\epsilon_{est}$ and $\hat{\epsilon}_{opt}$.

## C.2 STATISTICAL ERROR

In this section, we analyze the statistical error

$$
\epsilon_{est} := \sup_{\tau\in\mathcal{H},f\in\mathcal{F},u\in U}\left|J\left(\tau,u,f\right) - \widehat{J}\left(\tau,u,f\right)\right|.
$$

We mainly focus on the batch RL setting with *i.i.d.* samples $\mathcal{D} = [(x_i, x_i')]_{i=1}^N \sim \mathcal{T}_p(x, x')$, which has been studied by previous authors (e.g., Sutton et al., 2012; Nachum et al., 2019). However, as discussed in the literature (Antos et al., 2008; Lazaric et al., 2012; Dai et al., 2018; Nachum et al., 2019), using the blocking technique of Yu (1994), the statistical error provided here can be generalized to $\beta$-mixing samples in a single sample path. We omit this generalization for the sake of expositional simplicity.

To bound the $\epsilon_{est}$, we follow similar arguments by Dai et al. (2018); Nachum et al. (2019) via the covering number. For completeness, the definition is given below.

The Pollard's tail inequality bounds the maximum deviation via the covering number of a function class:

**Lemma 4 (Pollard (2012))** *Let $\mathcal{G}$ be a permissible class of $\mathcal{Z} \rightarrow [-M, M]$ functions and $\{Z_i\}_{i=1}^N$ are* i.i.d. *samples from some distribution. Then, for any given $\epsilon > 0$,*

$$\mathbb{P}\left(\sup_{g \in \mathcal{G}} \left| \frac{1}{N} \sum_{i=1}^N g(Z_i) - \mathbb{E}[g(Z)] \right| > \epsilon \right) \le 8\mathbb{E}\left[\mathcal{N}_1\left(\frac{\epsilon}{8}, \mathcal{G}, \{Z_i\}_{i=1}^N\right)\right] \exp\left(\frac{-N\epsilon^2}{512M^2}\right).$$

The covering number can then be bounded in terms of the function class's pseudo-dimension:

**Lemma 5 (Haussler (1995), Corollary 3)** *For any set $\mathcal{X}$, any points $x^{1:N} \in \mathcal{X}^N$, any class $\mathcal{F}$ of functions on $\mathcal{X}$ taking values in $[0, M]$ with pseudo-dimension $D_{\mathcal{F}} < \infty$, and any $\epsilon > 0$,*

$$\mathcal{N}_1\left(\epsilon, \mathcal{F}, x^{1:N}\right) \le e\left(D_{\mathcal{F}} + 1\right) \left(\frac{2eM}{\epsilon}\right)^{D_{\mathcal{F}}}.$$

The statistical error $\epsilon_{est}$ can be bounded using these lemmas.

**Lemma 6 (Stochastic error)** *Under the Assumption 2, if $\phi^*$ is $\kappa$-Lipschitz continuous and the psuedo-dimension of $\mathcal{H}$ and $\mathcal{F}$ are finite, with probability at least $1 - \delta$, we have*

$$\epsilon_{est} = \mathcal{O}\left(\sqrt{\frac{\log N + \log \frac{1}{\delta}}{N}}\right).$$

**Proof** The proof works by verifying the conditions in Lemma 4 and computing the covering number.

Denote the $h_{\tau,u,f}(x, x') = (1 - \gamma) f(x') + \gamma\tau(x) f(x') - \tau(x) \phi^*(f(x)) + \lambda u\tau(x) - \lambda u - \lambda\frac{1}{2}u^2$, we will apply Lemma 4 with $\mathcal{Z} = \Omega \times \Omega$, $Z_i = (x_i, x_i')$, and $\mathcal{G} = h_{\mathcal{H} \times \mathcal{F} \times U}$.

We check the boundedness of $h_{\zeta,u,f}(x, x')$. Based on Assumption 2, we only consider the $\tau \in \mathcal{H}$ and $u \in U$ bounded by $C$ and $C + 1$. We also rectify the $\|f\|_\infty \le C$. Then, we can bound the $\|h\|_\infty$:

$$\begin{aligned}
\|h_{\tau,u,f}\|_\infty &\le (1 + \|\tau\|_\infty) \|f\|_\infty + \|\tau\|_\infty \left(\max_{t \in [-C,C]} -\phi^*(t)\right) + \lambda C (\|\tau\|_\infty + 1) + \lambda C^2 \\
&\le (C + 1)^2 + C \cdot C_\phi + \lambda C (2C + 1) =: M,
\end{aligned}$$

where $C_\phi = \max_{t \in [-C,C]} -\phi^*(t)$. Thus, by Lemma 4, we have

$$\begin{aligned}
\mathbb{P}&\left(\sup_{\tau \in \mathcal{H}, f \in \mathcal{F}, u \in U} \left|\widehat{J}(\tau, u, f) - J(\tau, u, f)\right|\right) \\
&= \mathbb{P}\left(\sup_{\tau \in \mathcal{H}, f \in \mathcal{F}, u \in U} \left|\frac{1}{n} \sum_{i=1}^N h_{\zeta,u,f}(Z_i) - \mathbb{E}[h_{\zeta,u,f}]\right|\right) \\
&\le \mathbb{E}\left[\mathcal{N}_1\left(\frac{\epsilon}{8}, \mathcal{G}, \{Z_i\}_{i=1}^N\right)\right] \exp\left(\frac{-N\epsilon^2}{512M^2}\right). \quad (23)
\end{aligned}$$

Next, we check the covering number of $\mathcal{G}$. Firstly, we bound the distance in $\mathcal{G}$,

$$\frac{1}{N} \sum_{i=1}^{N} |h_{\tau_1, u_1, f_1}(Z_i) - h_{\tau_2, u_2, f_2}(Z_i)|$$

$$\leq \frac{C + C_\phi + \lambda(C+1)}{N} \sum_{i=1}^{N} |\tau_1(x_i) - \tau_2(x_i)| + \frac{1 + \gamma C}{N} \sum_{i=1}^{N} |f_1(x_i') - f_2(x_i')|$$

$$+ \frac{\kappa C}{N} \sum_{i=1}^{N} |f_1(x_i) - f_2(x_i)| + \lambda(2C+1)|u_1 - u_2|,$$

which leads to

$$\mathcal{N}_1 \left( (C_\phi + (3\lambda + 2 + \gamma + \kappa)(C+1)) \epsilon', \mathcal{G}, \{Z_i\}_{i=1}^{N} \right)$$

$$\leq \mathcal{N}_1 \left( \epsilon', \mathcal{H}, (x_i)_{i=1}^{N} \right) \mathcal{N}_1 \left( \epsilon', \mathcal{F}, (x_i')_{i=1}^{N} \right) \mathcal{N}_1 \left( \epsilon', \mathcal{F}, (x_i)_{i=1}^{N} \right) \mathcal{N}_1(\epsilon', U).$$

For the set $U = [-C-1, C+1]$, we have,

$$\mathcal{N}_1(\epsilon', U) \leq \frac{2C+2}{\epsilon'}.$$

Denote the pseudo-dimension of $\mathcal{H}$ and $\mathcal{F}$ as $D_{\mathcal{H}}$ and $D_{\mathcal{F}}$, respectively, we have

$$\mathcal{N}_1 \left( (C_\phi + (3 + 2\lambda + \kappa)(C+1)) \epsilon', \mathcal{G}, \{Z_i\}_{i=1}^{N} \right)$$

$$\leq e^3 (D_{\mathcal{H}} + 1)(D_{\mathcal{F}} + 1)^2 \left( \frac{2C+2}{\epsilon'} \right) \left( \frac{4eC}{\epsilon'} \right)^{D_{\mathcal{H}} + 2D_{\mathcal{F}}},$$

which implies

$$\mathcal{N}_1 \left( \frac{\epsilon}{8}, \mathcal{G}, \{Z_i\}_{i=1}^{N} \right)$$

$$\leq \frac{C+1}{2C} e^2 (D_{\mathcal{H}} + 1)(D_{\mathcal{F}} + 1)^2 \left( \frac{32(C_\phi + (3\lambda + 2 + \gamma + \kappa)(C+1))eC}{\epsilon} \right)^{D_{\mathcal{H}} + D_{\mathcal{F}} + 1}$$

$$= C_1 \left( \frac{1}{\epsilon} \right)^{D_1},$$

where $D_1 = D_{\mathcal{H}} + D_{\mathcal{F}} + 1$ and

$$C_1 = \frac{C+1}{2C} e^2 (D_{\mathcal{H}} + 1)(D_{\mathcal{F}} + 1)^2 (32(C_\phi + (3\lambda + 2 + \gamma + \kappa)(C+1))eC).$$

Combine this result with Equation 23, we obtain the bound for the statistical error:

$$\mathbb{P} \left( \sup_{\tau \in \mathcal{H}, f \in \mathcal{F}, u \in U} \left| \widehat{J}(\tau, u, f) - J(\tau, u, f) \right| \right) \leq 8C_1 \left( \frac{1}{\epsilon} \right)^{D_1} \exp \left( \frac{-N\epsilon^2}{512 M^2} \right). \tag{24}$$

Setting $\epsilon = \sqrt{\frac{C_2 \left( \log N + \log \frac{1}{\delta} \right)}{N}}$ with $C_2 = \max \left( (8C_1)^{\frac{2}{D_1}}, 512 M D_1, 512 M, 1 \right)$, we have

$$8C_1 \left( \frac{1}{\epsilon} \right)^{D_1} \exp \left( \frac{-N\epsilon^2}{512 M^2} \right) \leq \delta.$$

∎

### C.3 Optimization Error

In this section, we investigate the optimization error

$$\hat{\epsilon}_{opt} := \widehat{L}(\hat{\tau}) - \widehat{L}(\hat{\tau}_{\mathcal{H}}^*).$$

Notice our estimator $\min_{\tau \in \mathcal{H}} \max_{f \in \mathcal{F}, u \in U} \widehat{J}(\tau, u, f)$ is compatible with different parametrizations for $(\mathcal{H}, \mathcal{F})$ and different optimization algorithms, the optimization error will be different. For the general neural network for $(\tau, f)$, although there are several progress recently (Lin et al., 2018; Jin et al., 2019; Lin et al., 2019) about the convergence to a stationary point or local minimum, it remains a largely open problem to quantify the optimization error, which is out of the scope of this paper. Here, we mainly discuss the convergence rate with tabular, linear and kernel parametrization for $(\tau, f)$.

Particularly, we consider the linear parametrization particularly, *i.e.*, $\tau(x) = w_\tau^\top \psi_\tau(x)$ with $\{w_\tau, \psi_\tau(x) \geq 0\}$ and $f(x) = w_f^\top \psi_f(x)$. With such parametrization, the $\widehat{J}(\tau, u, f)$ is still convex-concave w.r.t $(w_\tau, w_f, u)$.

We can bound the $\hat{\epsilon}_{opt}$ by the primal-dual gap $\epsilon_{gap}$:

$$
\begin{aligned}
\hat{\epsilon}_{opt} &= \widehat{L}(\hat{\tau}) - \widehat{L}(\hat{\tau}_{\mathcal{H}}^*) \\
&\leq \max_{f \in \mathcal{F}, u \in U} \widehat{J}(\hat{\tau}, u, f) - \widehat{J}\left(\hat{\tau}_{\mathcal{H}}^*, \widehat{u}^*, \widehat{f}_{\mathcal{F}}^*\right) + \widehat{J}\left(\hat{\tau}_{\mathcal{H}}^*, \widehat{u}^*, \widehat{f}_{\mathcal{F}}^*\right) - \min_{\tau \in \mathcal{H}} \widehat{J}\left(\tau, \widehat{u}, \widehat{f}\right) \\
&= \underbrace{\max_{f \in \mathcal{F}, u \in U} \widehat{J}(\hat{\tau}, u, f) - \min_{\tau \in \mathcal{H}} \widehat{J}\left(\tau, \widehat{u}, \widehat{f}\right)}_{\epsilon_{gap}}.
\end{aligned}
$$

With vanilla SGD, we have $\epsilon_{gap} = \mathcal{O}\left(\frac{1}{\sqrt{T}}\right)$, where $T$ is the optimization steps (Nemirovski et al., 2009). Therefore, $\epsilon_{opt} = \mathbb{E}\left[\hat{\epsilon}_{opt}\right] = \mathcal{O}\left(\frac{1}{\sqrt{T}}\right)$, where the $\mathbb{E}\left[\cdot\right]$ is taken w.r.t. randomness in SGD.

## C.4 COMPLETE ERROR ANALYSIS

We are now ready to state the main theorm in a precise way:

**Theorem 3** *Under Assumptions 2 and 1, the stationary distribution $\mu$ exists,* i.e., $\max_{f \in \mathcal{F}^*} \mathbb{E}_{\mathcal{T} \circ \mu}[f] - \mathbb{E}_\mu[\phi^*(f)] = 0$. *If the $\phi^*(\cdot)$ is $\kappa$-Lipschitz continuous, $\|f\|_\infty \leq C < \infty$, $\forall f \in \mathcal{F}^*$, and the psuedo-dimension of $\mathcal{H}$ and $\mathcal{F}$ are finite, the error between the GenDICE estimate to $\tau^*(x) = \frac{u(x)}{p(x)}$ is bounded by*

$$
\mathbb{E}\left[J(\hat{\tau}) - J(\tau^*)\right] = \widetilde{\mathcal{O}}\left(\epsilon_{approx}(\mathcal{F}, \mathcal{H}) + \sqrt{\frac{1}{N}} + \epsilon_{opt}\right),
$$

*where $\mathbb{E}\left[\cdot\right]$ is w.r.t. the randomness in sample $\mathcal{D}$ and in the optimization algorithms. $\epsilon_{opt}$ is the optimization error, and $\epsilon_{approx}(\mathcal{F}, \mathcal{H})$ is the approximation induced by $(\mathcal{F}, \mathcal{H})$ for parametrization of $(\tau, f)$.*

**Proof** We have the total error as

$$
\mathbb{E}\left[J(\hat{\tau}) - J(\tau^*)\right] \leq 4\mathbb{E}\left[\epsilon_{est}\right] + \mathbb{E}\left[\epsilon_{opt}\right] + \epsilon_{approx}(\mathcal{F}, \mathcal{H}), \tag{25}
$$

where $\epsilon_{approx} := 2C_{\mathcal{T}, \kappa, \lambda}\epsilon_{approx}(\mathcal{F}) + C_{\phi, C, \lambda}\epsilon_{approx}(\mathcal{H})$. For $\epsilon_{opt}$, we can apply the results for SGD in Appendix C.3.

We can bound the $\mathbb{E}\left[\epsilon_{est}\right]$ by Lemma 6. Specifically, we have

$$
\mathbb{E}\left[\epsilon_{est}\right] = (1 - \delta)\sqrt{\frac{C_2\left(\log N + \log\frac{1}{\delta}\right)}{N}} + \delta M = \mathcal{O}\left(\sqrt{\frac{\log N}{N}}\right),
$$

by setting $\delta = \frac{1}{\sqrt{N}}$.

Plug all these bounds into Equation 25, we achieve the conclusion. $\blacksquare$

# D EXPERIMENTAL SETTINGS

## D.1 TABULAR CASE

For the Taxi domain, we follow the same protocol as used in Liu et al. (2018). The behavior and target policies are also taken from Liu et al. (2018) (referred in their work as the behavior policy for $\alpha = 0$). We use a similar taxi domain, where a grid size of $5 \times 5$ yields 2000 states in total ($25 \times 16 \times 5$, corresponding to 25 taxi locations, 16 passenger appearance status and 5 taxi status). We set our target policy as the final policy $\pi_*$ after running Q-learning (Sutton & Barto, 1998) for 1000 iterations, and

set another policy $\pi_+$ after 950 iterations as our base policy. The behavior policy is a mixture policy controlled by $\alpha$ as $\pi = (1-\alpha)\pi_* + \alpha\pi_+$, *i.e.*, the larger $\alpha$ is, the behavior policy is more close to the target policy. In this setting, we solve for the optimal stationary ratio $\tau$ exactly using matrix operations. Since Liu et al. (2018) perform a similar exact solve for $|S|$ variables $\mu(s)$, for better comparison we also perform our exact solve with respect to $|S|$ variables $\tau(s)$. Specifically, the final objective of importance sampling will require knowledge of the importance weights $\mu(a|s)/p(a|s)$.

For offline PageRank, the graph statistics are illustrated in Table 1, and the degree statistics and graph visualization are shown in Figure 8. For the BarabasiAlbert (BA) Graph, it begins with an initial connected network of $m_0$ nodes in the network. Each new node is connected to $m \leq m_0$ existing nodes with a probability that is proportional to the number of links

Table 1: Statistics of different graphs.

| Dataset | Number of Nodes | Number of Edges |
|---|---|---|
| BA (Small) | 100 | 400 |
| BA (Large) | 500 | 2000 |
| Cora | 2708 | 5429 |
| Citeseer | 3327 | 4731 |

that the existing nodes already have. Intuitively, heavily linked nodes ('hubs') tend to quickly accumulate even more links, while nodes with only a few links are unlikely to be chosen as the destination for a new link. The new nodes have a 'preference' to attach themselves to the already heavily linked nodes. For two real-world graphs, it is built upon the real-world citation networks. In our experiments, the weights of the BA graph is randomly drawn from a standard Gaussian distribution with normalization to ensure the property of the transition matrix. The offline data is collected by a random walker on the graph, which consists the initial state and next state in a single trajectory. In experiments, we vary the number of off-policy samples to validate the effectiveness of GenDICE with limited offline samples provided.

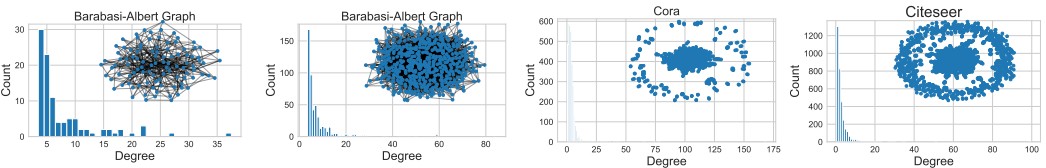

Figure 8: Degree statistics and visualization of different graphs.

## D.2 CONTINUOUS CASE

We use the Cartpole, Reacher and HalfCheetah tasks as given by OpenAI Gym. In importance sampling, we learn a neural network policy via behavior cloning, and use its probabilities for computing importance weights $\pi_*(a|s)/\pi(a|s)$. All neural networks are feed-forward with two hidden layers of dimension $64$ and $\tanh$ activations.

**Discrete Control Tasks** We modify the Cartpole task to be infinite horizon: We use the same dynamics as in the original task but change the reward to be $-1$ if the original task returns a termination (when the pole falls below some threshold) and $1$ otherwise. We train a policy on this task with standard Deep Q-Learning (Mnih et al., 2013) until convergence.

We then define the target policy $\pi_*$ as a weighted combination of this pre-trained policy (weight $0.7$) and a uniformly random policy (weight $0.3$). The behavior policy $\pi$ for a specific $0 \leq \alpha \leq 1$ is taken to be a weighted combination of the pre-trained policy (weight $0.55 + 0.15\alpha$) and a uniformly random policy (weight $0.45 - 0.15\alpha$). We train each stationary distribution correction estimation method using the Adam optimizer with batches of size $2048$ and learning rates chosen using a hyperparameter search from $\{0.0001, 0.0003, 0.001, 0.003\}$ and choose the best one as $0.0003$.

**Continuous Control Tasks** For the Reacher task, we train a deterministic policy until convergence via DDPG (Lillicrap et al., 2015). We define the target policy $\pi$ as a Gaussian with mean given by the pre-trained policy and standard deviation given by $0.1$. The behavior policy $\pi_b$ for a specific $0 \leq \alpha \leq 1$ is taken to be a Gaussian with mean given by the pre-trained policy and standard deviation given by $0.4 - 0.3\alpha$. We train each stationary distribution correction estimation method using the

Adam optimizer with batches of size 2048 and learning rates chosen using a hyperparameter search from {0.0001, 0.0003, 0.001, 0.003} and the optimal learning rate found was 0.003).

For the HalfCheetah task, we also train a deterministic policy until convergence via DDPG (Lillicrap et al., 2015). We define the target policy $\pi$ as a Gaussian with mean given by the pre-trained policy and standard deviation given by 0.1. The behavior policy $\pi_b$ for a specific $0 \leq \alpha \leq 1$ is taken to be a Gaussian with mean given by the pre-trained policy and standard deviation given by $0.2 - 0.1\alpha$. We train each stationary distribution correction estimation method using the Adam optimizer with batches of size 2048 and learning rates chosen using a hyperparameter search from {0.0001, 0.0003, 0.001, 0.003} and the optimal learning rate found was 0.003.

## E  ADDITIONAL EXPERIMENTS

### E.1  OPE FOR DISCRETE CONTROL

On the discrete control task, we modify the Cartpole task to be infinite horizon: the original dynamics is used but with a modified reward function: the agent will receive $-1$ if the environment returns a termination (*i.e.*, the pole falls below some threshold) and 1 otherwise. As shown in Figure 3, our method shows competitive results with IS and Model-Based in average reward case, but our proposed method finally outperforms these two methods in terms of log MSE loss. Specifically, it is relatively difficult to fit a policy with data collected by multiple policies, which renders the poor performance of IS.

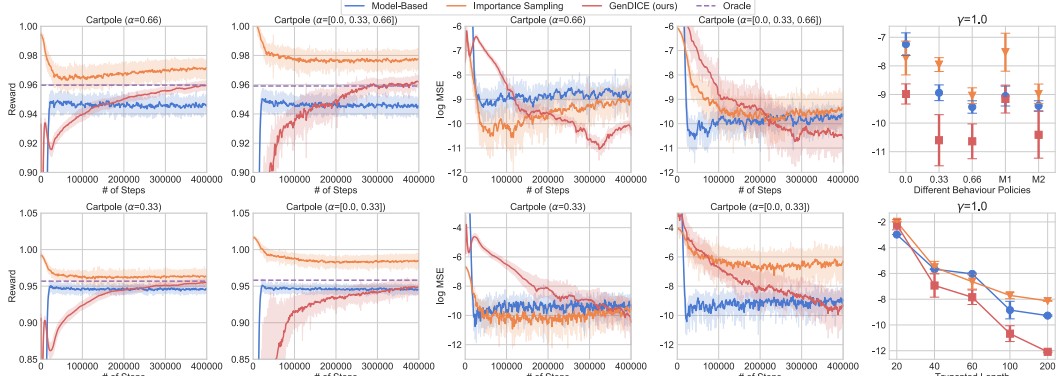

Figure 9: Results on Cartpole. Each plot in the first row shows the estimated average step reward over training and different behavior policies (higher $\alpha$ corresponds to a behavior policy closer to the target policy; the same in other figures); M1:$\alpha = [0.0, 0.33]$; M2: $\alpha = [0.0, 0.33, 0.66]$)

### E.2  ADDITIONAL RESULTS ON CONTINUOUS CONTROL

In this section, we show more results on the continuous control tasks, *i.e.*, HalfCheetah and Reacher. Figure 10 shows the log MSE towards training steps, and GenDICE outperforms other baselines with different behavior policies. Figure 11 better illustrates how our method beat other baselines, and can accurately estimate the reward of the target policy. Besides, Figure 12 shows GenDICE gives better reward estimation of the target policy. In these figures, the left three figures show the performance with off-policy dataset collected by single behavior policy from more difficult to easier tasks. The right two figures show the results, where off-policy dataset collected by multiple behavior policies.

Figure 13 shows the ablation study results in terms of estimated rewards. The left two figures shows the effects of different learning rate. When $\alpha = 0.33$, *i.e.*, the OPE tasks are relatively easier, GenDICE gets relatively good results in all learning rate settings. However, when $\alpha = 0.0$, *i.e.*, the estimation becomes more difficult, only GenDICE in larger learning rate gets reasonable estimation. Interestingly, we can see with larger learning rates, the performance becomes better, and when learning is 0.001 with $\alpha = 0.0$, the variance is very high, showing some cases the estimation becomes more accurate. The right three figures show different activation functions with different

behavior policy. The square and softplus function works well; while the exponential function shows poor performance under some settings. In practice, we use the square function since its low variance and better performance in most cases.

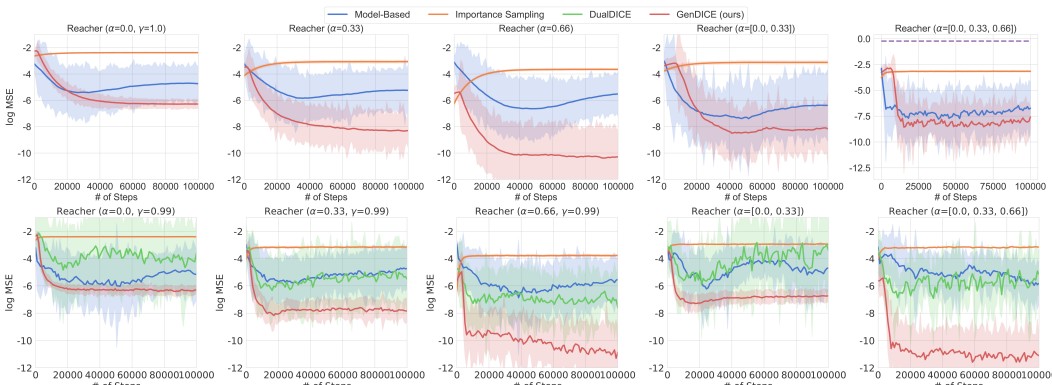

Figure 10: Results on Reacher. Each plot in the first row shows the estimated average step reward over training and different behavior policies (higher $\alpha$ corresponds to a behavior policy closer to the target policy; the same in other figures).

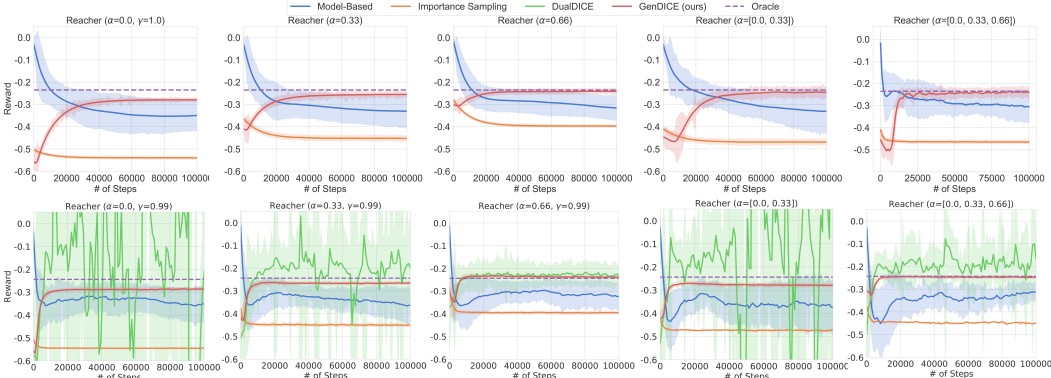

Figure 11: Results on Reacher. Each plot in the first row shows the estimated average step reward over training and different behavior policies (higher $\alpha$ corresponds to a behavior policy closer to the target policy; the same in other figures).

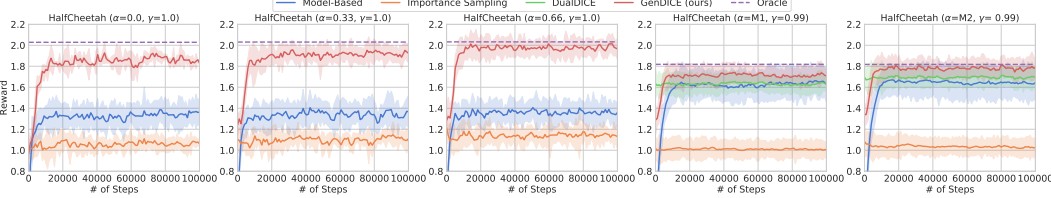

Figure 12: Results on HalfCheetah. Each plot in the first row shows the estimated average step reward over training and different behavior policies (higher $\alpha$ corresponds to a behavior policy closer to the target policy.

### E.3 COMPARISON WITH SELF-NORMALIZATION TRICK

The self-normalization trick used in (Liu et al., 2018) encodes the normalization constraint in $\tau$, while the principled optimization technique is considered in GenDICE. Further, the self-normalization trick will lead to several disadvantages theoretically, in both statistical and computational aspects :

Figure 13: Results of ablation study with different learning rates and activation functions. The plots show the estimated average step reward over training and different behavior policies .

**i)** It will generally not produce an unbiased solution. Although $\frac{1}{|\mathcal{D}|} \sum_{(s,a)\in\mathcal{D}} \tau(s, a)$ is an unbiased estimator for $\mathbb{E}[\tau]$, the plugin estimator $\frac{\tau(s,a)}{\frac{1}{|\mathcal{D}|} \sum_{(s,a)\in\mathcal{D}} \tau(s,a)}$ will be *biased* for $\frac{\tau(s,a)}{\mathbb{E}[\tau]}$.

**ii)** It will induce more computational cost. Specifically, the self-normalized ratio will be in the form of $\frac{\tau(s,a)}{\frac{1}{|\mathcal{D}|} \sum_{(s,a)\in\mathcal{D}} \tau(s,a)}$, which requires to go through all the samples in training set $\mathcal{D}$ even for just estimating one stochastic gradient, and thus, is prohibitive for large dataset.

Empirically, self-normalization is the most natural and the first idea we tried during this project. We have some empirical results about this method in the OPR setting.

Despite the additional computational cost, it performs worse than the proposed regularization technique used in the current version of Gen-DICE. Table 2 shows a comparison between self-normalization and regularization on OPR with $\chi^2$-divergence for BA graph with 100 nodes, 10,000 offline samples, and 20 trials. We stop the algorithms in the same running-time budget.

Table 2: Comparison between regularization and self-normalization.

|  | $\log$ **KL-divergence** |
| --- | --- |
| self-normalization | $-4.26 \pm 0.157$ |
| regularization | $-4.74 \pm 0.163$ |

