# OpenReview forum: "GenDICE: Generalized Offline Estimation of Stationary Values"
_ICLR.cc/2020/Conference — Accept (Talk)_

### Official Review · AnonReviewer3 · 2019-10-22
**Official Blind Review #3**

**Rating:** 8

**Review:**

Main contributions:
This paper generalizes the recent state-of-the-art behavior agnostic off-policy evaluation DualDice into a more general optimization framework: GenDice. Similar to DualDice, GenDice considers distribution correction over state, action pairs rather than state in Liu et al. (2018), which can handle behavior-agnostic settings. The optimization framework (in equation (9)) is novel and neat, and the practical algorithm seems more powerful than the previous DualDice. As a side product, it can also use to solve offline page rank problem.

Clarity:
This paper is well established and written.

Connection of theory and experiment:
I have a major concern for the theory 1 about the choice of regularizer $\lambda$. For infinite samples case, the derivation of theory 1 is reasonable since both term is nonnegative. However, in practice we will have empirical gap for the divergence term, thus picking a suitable $\lambda$ seems crucial for the experiment. I think a discussion on $\lambda$  for average case in experiment part should be added. And compared to Liu et al. (2018) which normalized the weight of $\tau$ in average case, which one is better in practice?

Overall I think this paper is good enough to be accepted by ICLR. The optimization framework can also inspire future algorithm using different divergence.

**Experience Assessment:**

I have published one or two papers in this area.

**Review Assessment: Checking Correctness Of Derivations And Theory:**

I carefully checked the derivations and theory.

**Review Assessment: Checking Correctness Of Experiments:**

I carefully checked the experiments.

**Review Assessment: Thoroughness In Paper Reading:**

I read the paper thoroughly.

---

> ### Author Response · Authors · 2019-11-15
> **Reply to reviewer #3**
>
> Thanks for the generally positive comments and constructive suggestions.  Below you can find the detailed response to the comments:
>
> 1. Effect of penalty weights in the empirical study:
>
> In our ablation study, we compared GenDICE in off-policy policy evaluation w/o the regularization term.  We have also added another ablation study to investigate the effect of the penalty weight in the offline PageRank task in Appendix E.3.  We vary $\lambda \in [0.1, 5]$ with the $\chi^2$-divergence in the GenDICE estimator.
>
> Within a large range of $\lambda$, the performance of GenDICE is quite consistent, which justifies Theorem 1. The penalty multiplies with $\lambda$. Therefore, with increasing $\lambda$, the variance of the stochastic gradient estimator also increases, which explains the increasing variance for a large $\lambda$ in Fig.13(b). In practice, we found that $\lambda = 1$ is a reasonable choice for general cases.
>
> 2. Comparison to Liu et al. (2018):
>
> We added a comparison to Liu et al. (2018) in the Taxi domain (both discounted and average cases) using their implementation with a learned behavior policy. As we can see in Fig. 2, the proposed GenDICE performs the best among all the competitors.
>
> Best,
> Authors

---

> > ### Comment · AnonReviewer3 · 2019-11-15
> > **Thank you for the response**
> >
> > Thank you for the response, the ablation study is clear and provide important guidance in empirical experiments.
> >
> > For the comparison to Liu et al. normalized method, it seems that is only used in continuous setting. For tabular setting since we can constraint the summation of density function to be 1, a constraint optimization (like mirror descend or quadratic programming is enough). The normalized trick in their equation (11) (divided by $z_w$) I believe is mainly used in continuous setting.
> >
> > I am still wondering compared to the regularizer you use in equation (11), which one is more stable in optimization process with neural network?

---

### Official Review · AnonReviewer1 · 2019-10-23
**Official Blind Review #1**

**Rating:** 8

**Review:**

In this paper the authors proposed a framework for off-policy value estimation under the scenario of infinite horizon RL tasks. The new proposed method utilize the variational representation of $f$-divergence, which quantifies the difference between $\mathcal{T}\tau$ and $\tau p$, where $\tau$ is the parametric density ratio between the unknown behavior policy data and the target policy. If only if $\tau$ is the true density ratio, the loss $\mathcal{D}_{f}(\mathcal{T}\tau || \tau p) = 0$.

Compared with prior work (Nachum et. al 2019), the new proposed framework can generalize the undiscounted case $\gamma = 1$, and the derivation for the new algorithm is quite simple and easy to follow. The experimental results show the advantage of the proposed methods over baseline methods such as model-based, DualDice etc, for both discrete and continuous cases. Moreover, I have two specific questions:

- The choice of $f$-divergence. Although the author mentioned the difficulty of using the dual representation of KL divergence, it would be nice to have an ablation study that shows the effectiveness of various $f$-divergence (Personally I think Jensen-Shannon Divergence may be also a good choice).

-The authors should also have a discussion that similar idea can be generalized to more general  distribution metrics such as Integral Probability Metrics, specifically wasserstein-1 distance (similar to wasserstein-gan) or maximum mean discrepancy (Maybe it is unnecessary to conduct experiments, some discussion should be enough to clarify the relationship. I think there is a concurrent submission using MMD metrics).


Overall I think this is a good paper and I recommend for acceptance.

Reference Papers:
- Nowozin, Sebastian, Botond Cseke, and Ryota Tomioka. "f-gan: Training generative neural samplers using variational divergence minimization." Advances in neural information processing systems. 2016.
- Arjovsky, Martin, Soumith Chintala, and Léon Bottou. "Wasserstein gan." arXiv preprint arXiv:1701.07875 (2017).
- Nachum, Ofir, et al. "DualDICE: Behavior-Agnostic Estimation of Discounted Stationary Distribution Corrections." arXiv preprint arXiv:1906.04733 (2019).
- Anonymous, “Black-box Off-policy Estimation for Infinite-Horizon Reinforcement Learning”, submitted to ICLR 2020.


**Experience Assessment:**

I have published one or two papers in this area.

**Review Assessment: Checking Correctness Of Derivations And Theory:**

I carefully checked the derivations and theory.

**Review Assessment: Checking Correctness Of Experiments:**

I assessed the sensibility of the experiments.

**Review Assessment: Thoroughness In Paper Reading:**

I read the paper thoroughly.

---

> ### Author Response · Authors · 2019-11-15
> **Reply to reviewer #1**
>
> Thanks for the insightful comments and constructive suggestions. In the revision, we have added the suggested discussion and an ablation study that investigates the choice of divergences. The summary is listed below:
>
> 1. The generalized integral probability metrics (IPM)
>
> The proposed estimator is compatible with IPM, e.g., MMD, Wasserstein-$1$ distance, and Dudley metric, as discussed in the Remark above section 3.4. We have further elaborated on this point in the revision.
>
> By definition, the IPM naturally leads to a similar min-max optimization with identity function as $\phi^*(\cdot)$ and different feasible sets of the dual functions $f$:
>
>     i) MMD requires $f$ to be in an RKHS and its RKHS norm should be smaller than $1$, i.e., $\|f\|_{\mathcal{H}}\le 1$;
>     ii) Wasserstein-$1$ distance requires $f$ to be $1$-Lipschitz, i.e., $\|\nabla f\|_2\le 1$;
>     iii) Dudley metric requires $\|f\|_{BL}\le 1$ with $\|f\|_{BL}:= \|f\|_\infty + \|\nabla f\|_2$.
>
> These requirements on the dual function may incur some additional difficulty in practice. For the Wasserstein-$1$ distance and Dudley metric, we might need the extra gradient penalty, which requires extra computation to take the gradient through a gradient. Meanwhile, the consistency of the surrogate loss under regularization is not clear. For MMD, we can obtain the closed-form solution for the dual function, which saves the cost of the inner optimization with the trade-off of requiring two independent samples in each update for outer optimization. Moreover, it relies on the condition that the dual function must be in some RKHS, which introduces extra important kernel parameters to be tuned, and in practice might not be sufficiently flexible compared to neural networks.
>
> Thanks for pointing out the concurrent submission. Both aim to solve for the stationary distribution. However,  i) we explicitly handle the degeneracy issue of the naive estimator in a principled way, leading to an easier optimization problem;  and ii) the proposed estimator is more general, which is compatible with different divergences.  We will discuss this concurrent submission in our final version.
>
>
> 2. The empirical performance using different divergences
>
> We tested GenDICE with several alternative divergences on the offline PageRank task in Appendix E.3, including Wasserstein-$1$ distance, Jensen-Shannon divergence, $KL$-divergence, Hellinger divergence, and MMD. To ensure that the dual function is $1$-Lipchitz, we added the gradient penalty. We used a learned Gaussian kernel for MMD.
>
> As we can see in Fig. 13(a), with these different divergences, the proposed GenDICE estimator can always achieve significantly better performance compared to the model-based estimator, showing that the GenDICE estimator is compatible with many different divergences. Most of the divergences, with appropriate techniques added to handle the optimization difficulties and careful tuning of the additional parameters, can achieve similar performance, consistent with the phenomena observed with variants of GANs [1]. However, $KL$-divergence is an outlier, performing noticeably worse, which might be caused by the ill-behaved $\exp(\cdot)$ in its conjugate function. The $\chi^2$-divergence and JS-divergence are better, both achieving good performance with fewer parameters to tune.
>
>
> [1] Mario Lucic, Karol Kurach, Marcin Michalski, Sylvain Gelly, and Olivier Bousquet. Are GANs created equal? a large-scale study. In Advances in neural information processing systems, pp. 700–709, 2018.
>
>
> Best,
> Authors

---

> > ### Comment · AnonReviewer1 · 2019-11-15
> > **Thank you for the Reponse**
> >
> > Thank the author for the detailed response. The ablation study answers all my questions and the current paper is quite solid.  It would be great if the author can open source the code, which will definitely benefit the OPE community.

---

### Official Review · AnonReviewer2 · 2019-10-23
**Official Blind Review #2**

**Rating:** 8

**Review:**

This paper proposes a new estimator to infer the stationary distribution of a Markov chain, with data from another Markov chain. The method estimates the ratio between stationary distribution of target MC and the empirical data distribution.  It is based on the observation that the ratio is a fixed point solution to certain operators. The proposed method could work in the behavior-agnostic and undiscounted case, which is unsolved by the previous method.

This paper tackles an interesting problem with an increasing number of studies in the reinforcement learning community and gives a practical algorithm with strong empirical justification, as well as theoretical justification. I think this paper should be accepted.

Detailed comments:
1) This paper provides experiment results in multiple domains, including two continuous control domains which are more complex than experiments in previous OPE methods. The paper also provides many details about the learning dynamics and ablation studies, which is very useful for the reader to understand the result of the paper.
2) The theoretical result is as same strong as previous work DICE and DualDICE, under similar assumptions.
3) I appreciate this paper formalizes the two difficulties of degeneration and intractability, and then explain how those are addressed in a principled way. Degeneration is important and is at least ignored in two similar works on this topic.

**Experience Assessment:**

I have published one or two papers in this area.

**Review Assessment: Checking Correctness Of Derivations And Theory:**

I assessed the sensibility of the derivations and theory.

**Review Assessment: Checking Correctness Of Experiments:**

I assessed the sensibility of the experiments.

**Review Assessment: Thoroughness In Paper Reading:**

I read the paper at least twice and used my best judgement in assessing the paper.

---

> ### Author Response · Authors · 2019-11-15
> **Reply to reviewer #2**
>
> Thanks for the encouraging comments.  We will keep improving the draft. We have refined the paper as listed above in the summary of revisions.
>
> Best,
> Authors

---

### Author Response · Authors · 2019-11-15
**Summary of Revisions**

We would like to thank all the reviewers for taking the time to contribute their insightful comments, which helped us improve the paper. Our detailed point-to-point responses can be found below, and we have also carefully updated the manuscript to follow the constructive suggestions from the reviewers.

Here is a brief summary of major updates made to the revised manuscript:
1. Added an ablation study on the choice of the divergence: Wasserstein-$1$ distance, Jensen-Shannon divergence, $KL$-divergence, Hellinger divergence, and MMD.
2. Added an ablation study on the effect of the constraint weight in practice.
3. Added more discussion about the choice of divergence.

We will further improve the paper based on the reviewers' comments.

---

### Decision · Program_Chairs · 2019-12-19

**Decision:**

Accept (Talk)

**Comment:**

The authors develop a framework for off-policy value estimation for infinite horizon RL tasks, for estimating the stationary distribution of a Markov chain. Reviewers were uniformly impressed by the work, and satisfied by the author response. Congratulations!